# Graph Sparsification via Mixture of Graphs

**Guibin Zhang**[1*], **Xiangguo Sun**[2*], **Yanwei Yue**[1*], **Chonghe Jiang**[2],
**Kun Wang**[3†], **Tianlong Chen**[4†], **Shirui Pan**[5]
[1]Tongji University  [2]CUHK  [3]NTU  [4]UNC-Chapel Hill  [5]Griffith University

## Abstract

Graph Neural Networks (GNNs) have demonstrated superior performance across various graph learning tasks but face significant computational challenges when applied to large-scale graphs. One effective approach to mitigate these challenges is graph sparsification, which involves removing non-essential edges to reduce computational overhead. However, previous graph sparsification methods often rely on a single global sparsity setting and uniform pruning criteria, failing to provide customized sparsification schemes for each node's complex local context. In this paper, we introduce Mixture-of-Graphs (MoG), leveraging the concept of Mixture-of-Experts (MoE), to dynamically select tailored pruning solutions for each node. Specifically, MoG incorporates multiple sparsifier experts, each characterized by unique sparsity levels and pruning criteria, and selects the appropriate experts for each node. Subsequently, MoG performs a mixture of the sparse graphs produced by different experts on the Grassmann manifold to derive an optimal sparse graph. One notable property of MoG is its entirely local nature, as it depends on the specific circumstances of each individual node. Extensive experiments on four large-scale OGB datasets and two superpixel datasets, equipped with five GNN backbones, demonstrate that MoG (I) identifies subgraphs at higher sparsity levels ($8.67\% \sim 50.85\%$), with performance equal to or better than the dense graph, (II) achieves $1.47-2.62\times$ speedup in GNN inference with negligible performance drop, and (III) boosts "top-student" GNN performance ($1.02\% \uparrow$ on RevGNN+OGBN-PROTEINS and $1.74\% \uparrow$ on DeeperGCN+OGBG-PPA). The source code is available at https://github.com/yanweiyue/MoG.

## 1 Introduction

Graph Neural Networks (GNNs) (Sun et al., 2023a; Zhou et al., 2020) have become prominent for confronting graph-related learning tasks, including social recommendation (Wu et al., 2021; Yu et al., 2022), fraud detection (Sun et al., 2022; Wang et al., 2019a; Cheng et al., 2020), drug design (Zhang & Liu, 2023), and many others (Wu et al., 2023; Sun et al., 2023b). The superiority of GNNs stems from iterative *aggregation* and *update* processes. The former accumulates embeddings from neighboring nodes via sparse matrix-based operations (*e.g.*, sparse-dense matrix multiplication (`SpMM`) and sampled dense-dense matrix multiplication (`SDDMM`) (Fey & Lenssen, 2019; Wang et al., 2019b)), and the latter updates the central nodes' embeddings using dense matrix-based operations (*e.g.*, `MatMul`) (Fey & Lenssen, 2019; Wang et al., 2019b). `SpMM` typically contributes the most substantial part ($\sim70\%$) to the computational demands (Liu et al., 2023b; Zhang et al., 2024b), influenced largely by the graph's scale. Nevertheless, large-scale graphs are widespread in real-world scenarios (Wang et al., 2022a; Jin et al., 2021; Zhang et al., 2024a), leading to substantial computational burdens, which hinder the efficient processing of features during the training and inference, posing headache barriers to deploying GNNs in the limited resources environments.

To conquer the above challenge, graph sparsification (Chen et al., 2023; Hashemi et al., 2024) has recently seen a revival as it directly reduces the *aggregation* process associated with `SpMM` (Liu et al., 2023b; Zhang et al., 2024b) in GNNs. Specifically, graph sparsification is a technique that approximates a given graph by creating a sparse subgraph with a subset of vertices and/or edges. Since the execution time of `SpMM` is directly related to the number of edges in the graph, this method can significantly accelerate GNN training or inference. Existing efforts such as UGS (Chen et al.,

---

*Equal Contribution,   †Corresponding Authors

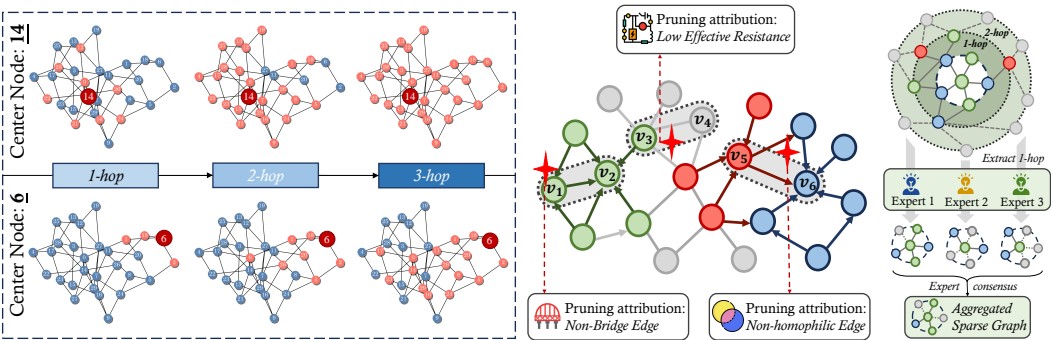

Figure 1: (*Left*) We illustrated the $k$-hop neighborhood expansion rates for nodes 6 and 14, which is proportional to the amount of message they receive as the GNN layers deepen; (*Middle*) The local patterns of different nodes vary, hence the attributions of edge pruning may also differ. For instance, pruning $(v_1, v_2)$ might be due to its non-bridge identity, while pruning $(v_5, v_6)$ could be attributed to its non-homophilic nature; (*Right*) The overview of our proposed MoG.

2021), DSpar (Liu et al., 2023b), and AdaGLT (Zhang et al., 2023) have achieved notable successes, with some maintaining GNN performance even with up to 40% edge sparsity.

Beyond serving as a **computational accelerator**, the purpose of graph sparsification extends further. Another research line leverages graph sparsification as a **performance booster** to remove task-irrelevant edges and pursue highly performant and robust GNNs (Zheng et al., 2020). Specifically, it is argued that due to uncertainty and complexity in data collection, graph structures are inevitably redundant, biased, and noisy (Li et al., 2024). Therefore, employing graph sparsification can effectively facilitate the evolution of graph structures towards cleaner conditions (Zheng et al., 2020; Luo et al., 2021), and finally boost GNN performance.

However, existing sparsification methods, namely *sparsifiers*, whether aimed at achieving higher sparsity or seeking enhanced performance, often adopt a rigid, global approach to conduct graph sparsification, thus suffering from the inflexibility in two aspects:

❶ *Inflexibility of sparsity level.* Previous sparsifiers globally score all edges uniformly and prune them based on a preset sparsity level (Chen et al., 2023). However, as shown in Figure 1 (*Left*), the degrees of different nodes vary, which leads to varying rates of $k$-hop neighborhood expansion. This phenomenon, along with prior work on node-wise aggregation (Lai et al., 2020; Wang et al., 2023a), suggests that *different nodes require customized sparsity levels tailored to their specific connectivity and local patterns.*

❷ *Inflexibility of sparsity criteria.* Previous sparsifiers often operate under a unified guiding principle, such as pruning non-bridge edges (Wang et al., 2022b), non-homophilic edges (Gong et al., 2023), or edges with low effective resistance (Spielman & Srivastava, 2008; Liu et al., 2023b), among others. However, as illustrated in Figure 1 (*Middle*), the context of different nodes varies significantly, leading to varied rationales for edge pruning. Therefore, it is essential to *select appropriate pruning criteria tailored to the specific circumstances of each node to customize the pruning process effectively.*

Based on these observations and reflections, we propose the following challenge: *Can we customize the sparsity level and pruning criteria for each node, in the meanwhile ensuring the efficiency of graph sparsification?* Towards this end, we propose a novel graph sparsification method dubbed Mixture of Graphs (**MoG**). It comprises multiple *sparsifier experts*, each equipped with distinct pruning criteria and sparsity settings, as in Figure 1 (*Right*). Throughout the training process, MoG dynamically selects the most suitable sparsifier expert for each node based on its neighborhood properties. This fosters specialization within each MoG expert, focusing on specific subsets of nodes with similar neighborhood contexts. After each selected expert prunes the 1-hop subgraph of the central nodes and outputs its sparse version, MoG seamlessly integrates these sparse subgraphs on the Grassmann manifold in an expert-weighted manner, thereby forming an optimized sparse graph.

We validate the effectiveness of MoG through a comprehensive series of large-scale tasks. Experiments conducted across six datasets and three GNN backbones showcase that MoG can ❶ effectively locate well-performing sparse graphs, maintaining GNN performance losslessly at satisfactory graph

sparsity levels ($8.67\% \sim 50.85\%$), and even only experiencing a $1.65\%$ accuracy drop at $69.13\%$ sparsity on OGBN-PROTEINS; ❷ achieve a tangible $1.47 \sim 2.62\times$ inference speedup with negligible performance drop; and ❸ boost ROC-AUC by $1.81\%$ on OGBG-MOLHIV, $1.02\%$ on ogbn-proteins and enhances accuracy by $0.95\%$ on OGBN-ARXIV compared to the vanilla backbones.

The key contributions of MoG can be found in appendix H.2.

## 2  TECHNICAL BACKGROUND

**Notations & Problem Formulation**  We consider an undirected graph $\mathcal{G} = \{\mathcal{V}, \mathcal{E}\}$, with $\mathcal{V}$ as the node set and $\mathcal{E}$ the edge set. The node features of $\mathcal{G}$ is represented as $\mathbf{X} \in \mathbb{R}^{N \times F}$, where $N = |\mathcal{V}|$ signifies the total number of nodes in the graph. The feature vector for each node $v_i \in \mathcal{V}$, with $F$ dimensions, is denoted by $x_i = \mathbf{X}[i, \cdot]$. An adjacency matrix $\mathbf{A} \in \{0, 1\}^{N \times N}$ is utilized to depict the inter-node connectivity, where $\mathbf{A}[i, j] = 1$ indicates an edge $e_{ij} \in \mathcal{E}$, and $0$ otherwise. For our task of graph sparsification, the core objective is to identify a subgraph $\mathcal{G}_{\text{sub}}$ given a sparsity ratio $s\%$:

$$\mathcal{G}^{\text{sub}} = \{\mathcal{V}, \mathcal{E} \setminus \mathcal{E}'\}, \ s\% = \frac{|\mathcal{E}'|}{|\mathcal{E}|}, \tag{1}$$

where $\mathcal{G}^{\text{sub}}$ only modifies the edge set $\mathcal{E}$ without altering the node set $\mathcal{V}$, and $\mathcal{E}'$ denotes the removed edges, and $s\%$ represents the ratio of removed edges.

**Graph Neural Networks**  Graph neural networks (GNNs) (Wu et al., 2020) have become pivotal for learning graph representations, achieving benchmark performances in various graph tasks at node-level (Xiao et al., 2022), edge-level (Sun et al., 2021), and graph-level (Liu et al., 2022a). At the node-level, two of the most famous frameworks are GCN (Kipf & Welling, 2017) and GraphSAGE (Hamilton et al., 2017), which leverages the message-passing neural network (MPNN) framework (Gilmer et al., 2017) to aggregate and update node information iteratively. For edge-level and graph-level tasks, GCN and GraphSAGE can be adapted by simply incorporating a predictor head or pooling layers. Nevertheless, there are still specialized frameworks like SEAL (Zhang & Chen, 2018) and Neo-GNN (Yun et al., 2021) for link prediction, and DiffPool (Ying et al., 2018) and PNA (Corso et al., 2020) for graph classification. Regardless of the task, MPNN-style GNNs generally adhere to the following paradigm:

$$\mathbf{h}_i^{(l)} = \text{COMB}\left(\mathbf{h}_i^{(l-1)}, \text{AGGR}\{\mathbf{h}_j^{(k-1)} : v_j \in \mathcal{N}(v_i)\}\right), \ 0 \leq l \leq L \tag{2}$$

where $L$ is the number of GNN layers, $\mathbf{h}_i^{(0)} = \mathbf{x}_i$, and $\mathbf{h}_i^{(l)}(1 \leq l \leq L)$ denotes $v_i$'s node embedding at the $l$-th layer. $\text{AGGR}(\cdot)$ and $\text{COMB}(\cdot)$ represent functions used for aggregating neighborhood information and combining ego- and neighbor-representations, respectively.

**Graph Sparsification**  Graph sparsification methods can be categorized by their utility into two main types: computational accelerators and performance boosters. Regarding computational accelerators, early works aimed at speeding up traditional tasks like graph partitioning/clustering often provide theoretical assurances for specific graph properties, such as pairwise distances (Althöfer et al., 1990), cuts (Abboud et al., 2022), eigenvalue distribution (Batson et al., 2013), and effective resistance (Spielman & Srivastava, 2008). More contemporary efforts focus on the GNN training and/or inference acceleration, including methods like SGCN (Li et al., 2020b), GEBT (You et al., 2022), UGS (Chen et al., 2021), DSpar (Liu et al., 2023b), and AdaGLT (Zhang et al., 2024a). Regarding performance boosters, methods like NeuralSparse (Zheng et al., 2020) and PTDNet (Luo et al., 2021) utilize parameterized denoising networks to eliminate task-irrelevant edges. SUBLIME (Liu et al., 2022b) and Nodeformer (Wu et al., 2022) also involve refining or inferring a cleaner graph structure followed by $k$-nearest neighbors ($k$NN) sparsification.

**Mixture of Experts**  The Mixture of Experts (MoE) concept (Jacobs et al., 1991) traces its origins to several seminal works (Chen et al., 1999; Jordan & Jacobs, 1994). Recently, the sparse MoE architecture (Shazeer et al., 2017; Lepikhin et al., 2020; Fedus et al., 2021; Clark et al., 2022) has regained attention due to its capacity to support the creation of vast (language) models with trillions of parameters (Clark et al., 2022; Hoffmann et al., 2022). Given its stability and generalizability, sparse MoE is now broadly implemented in modern frameworks across various domains, including vision (Riquelme et al., 2021), multi-modal (Mustafa et al., 2022), and multi-task learning (Ma et al.,

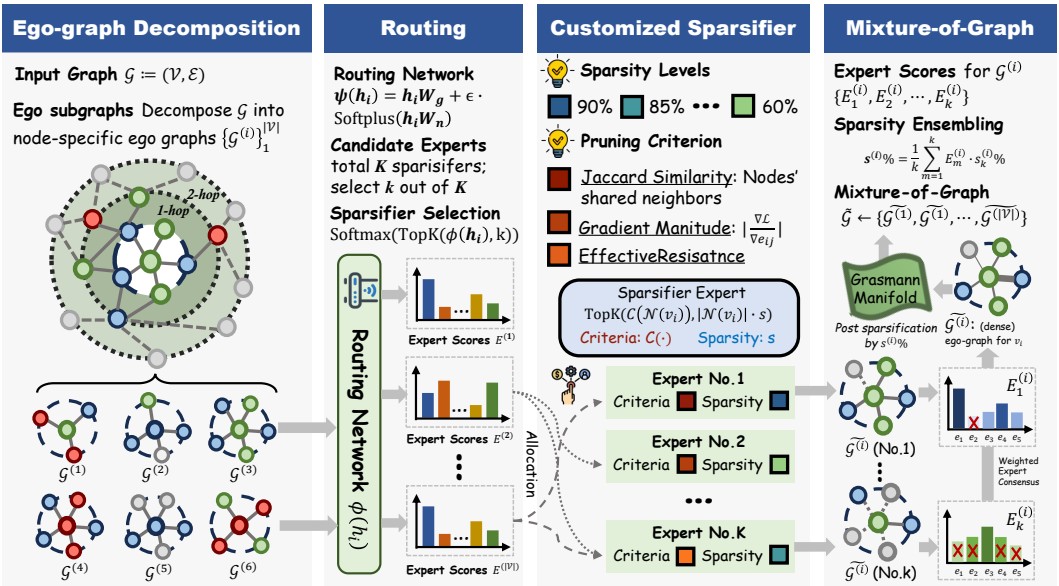

Figure 2: The overview of our proposed method. MoG primarily comprises ego-graph decomposition, expert routing, expert customization, and the final graph mixture. For simplicity, we only showcase three pruning criteria including Jaccard similarity, gradient magnitude, and effective resistance.

2018; Zhu et al., 2022). As for graph learning, MoE has been explored for applications in graph classification (Hu et al., 2022), scene graph generation (Zhou et al., 2022), molecular representation (Kim et al., 2023), graph fairness (Liu et al., 2023a), and graph diversity modeling (Wang et al., 2024).

## 3 METHODOLOGY

### 3.1 OVERVIEW

Figure 2 illustrates the workflow of our proposed MoG. Specifically, for an input graph, MoG first decomposes it into 1-hop ego graphs for each node. For each node and its corresponding ego graph, a routing network calculates the expert scores. Based on the router's decisions, sparsifier experts with different sparsity levels and pruning criteria are allocated to different nodes. Ultimately, a mixture of graphs is obtained based on the weighted consensus of the sparsifier experts. In the following sections, we will first detail how to route different experts in Section 3.2, then describe how to explicitly model various sparsifier experts in Section 3.3 and how to ensemble the sparse graphs output by experts on the Grassmann manifold in Section 3.4. Finally, the overall optimization process and complexity analysis of MoG is placed in Section 3.5.

### 3.2 ROUTING TO DIVERSE EXPERTS

Following the classic concept of a (sparsely-gated) mixture-of-experts (Zhao et al., 2024), which assigns the most suitable expert(s) to each input sample, MoG aims to allocate the most appropriate sparsity level and pruning criteria to each input node. To achieve this, we first decompose the input graph $\mathcal{G} = \{\mathcal{V}, \mathcal{E}\}$ into 1-hop ego graphs centered on different nodes, denoted as $\{\mathcal{G}^{(1)}, \mathcal{G}^{(2)}, \cdots, \mathcal{G}^{(N)}\}$, where $\mathcal{G}^{(i)} = \{\mathcal{V}^{(i)}, \mathcal{E}^{(i)}\}$, $\mathcal{V}^{(i)} = \{v_j | v_j \in \mathcal{N}(v_i)\}$, $\mathcal{E}^{(i)} = \{e_{ij} | (v_i, v_j) \in \mathcal{E}\}$. Assuming we have $K$ sparsifier experts, for each node $v_i$ and its corresponding ego graph $\mathcal{G}^{(i)}$, we aim to select $k$ most suitable experts. We employ the noisy top-$k$ gating mechanism following Shazeer et al. (2017):

$$\Psi(\mathcal{G}^{(i)}) = \text{Softmax}(\text{TopK}(\psi(x_i), k)), \tag{3}$$

$$\psi(x_i) = x_i W_g + \epsilon \cdot \text{Softplus}(x_i W_n), \tag{4}$$

where $\psi(x_i) \in \mathbb{R}^K$ is the calculated scores of $v_i$ for total $K$ experts, $\text{TopK}(\cdot)$ is a selection function that outputs the largest $k$ values, and $\Psi(\mathcal{G}^{(i)}) \in \mathbb{R}^k = [E_1^{(i)}, E_2^{(i)}, \cdots, E_k^{(i)}]$ represents those for

selected $k$ experts. In $\Psi(\mathcal{G}^{(i)})$, $\epsilon \in \mathcal{N}(0, 1)$ denotes the standard Gaussian noise, $W_g \in \mathbb{R}^{K \times F}$ and $W_n \in \mathbb{R}^{K \times F}$ are trainable parameters that learn clean and noisy scores, respectively.

After determining the appropriate experts, we proceed to generate different sparse graphs with diverse sparsifiers. We denote each sparsifier by $\kappa(\cdot)$, which takes in a dense graph $\mathcal{G}$ and outputs a sparse one $\tilde{\mathcal{G}} = \kappa(\mathcal{G})$. Based on this, for each node $v_i$ and its ego graph $\mathcal{G}^{(i)}$, the routing network selects $k$ experts that produce $k$ sparse ego graphs. Notably, sparsifiers differ in their pruning rates (*i.e.* the proportion of the edges to be removed) and the pruning criteria, which will be detailed in Section 3.3. MoG's dynamic selection of different sparsifiers for each node aids in identifying pruning strategies truly adapted to the node's local context. Formally, the mixture of $k$ sparse graphs can be written as:

$$\widehat{\mathcal{G}^{(i)}} = \text{ESMB}(\{\widetilde{\mathcal{G}}_m^{(i)}\}_{m=1}^k), \quad \widetilde{\mathcal{G}}_m^{(i)} = \kappa^m(\mathcal{G}^{(i)}), \tag{5}$$

where $\text{ESM}(\cdot)$ is a combination function that receives $k$ sparse graphs and ideally outputs an emsemble version $\widehat{\mathcal{G}^{(i)}} = \{\widehat{\mathcal{V}^{(i)}}, \widehat{\mathcal{E}^{(i)}}\}$ that preserves their desirable properties. It is noteworthy that, MoG can *seamlessly* integrate with any GNN backbone after obtaining each node's sparse ego graph. Specifically, we modify the aggregation method in Equation (2) as follows:

$$\mathbf{h}_i^{(l)} = \text{COMB}\left(\mathbf{h}_i^{(l-1)}, \text{AGGR}\{\mathbf{h}_j^{(k-1)} : v_j \in \widehat{\mathcal{V}^{(i)}}\}\right). \tag{6}$$

MoG acts as a plug-and-play module that can be pre-attached to any GNN architecture, leveraging multi-expert sparsification to enhance GNNs with (1) performance improvements from removing task-irrelevant edges (validated in Section 4.3); (2) resistance to high graph sparsity through precise and customized sparsification (validated in Section 4.2). The remaining questions now are: *how can we design explicitly different sparsifiers?* and further, *how can we develop an effective combination function that integrates the sparse graphs from different experts?*

## 3.3 CUSTOMIZED SPARSIFIER MODELING

With the workflow of MoG in mind, in this section, we will delve into how to design sparsifiers driven by various pruning criteria and different levels of sparsity. Revisiting graph-related learning tasks, their objective can generally be considered as learning $P(\mathbf{Y}|\mathcal{G})$, which means learning the distribution of the target $\mathbf{Y}$ given an input graph. Based on this, a sparsifier $\kappa(\cdot)$ can be formally expressed as follows:

$$P(\mathbf{Y}|\mathcal{G}) \approx \sum_{g \in \mathbb{S}_\mathcal{G}} P(\mathbf{Y} \mid \tilde{\mathcal{G}}) P(\tilde{\mathcal{G}} \mid \mathcal{G}) \quad \approx \sum_{g \in \mathbb{S}_\mathcal{G}} Q_\Theta(\mathbf{Y} \mid \tilde{\mathcal{G}}) Q_\kappa(\tilde{\mathcal{G}} \mid \mathcal{G}) \tag{7}$$

where $\mathbb{S}_\mathcal{G}$ is a class of sparsified subgraphs of $\mathcal{G}$. The second term in Equation (7) aims to approximate the distribution of $\mathbf{Y}$ using the sparsified graph $\tilde{\mathcal{G}}$ as a bottleneck, while the third term uses two approximation functions $Q_\Theta$ and $Q_\kappa$ for $P(\mathbf{Y} \mid \tilde{\mathcal{G}})$ and $P(\tilde{\mathcal{G}} \mid \mathcal{G})$ parameterized by $\Theta$ and $\kappa$ respectively. The parameter $\Theta$ typically refers to the parameters of the GNN, while the sparsifier $\kappa(\cdot)$, on the other hand, is crafted to take an ego graph $\mathcal{G}^{(i)}$ and output its sparsified version $\tilde{\mathcal{G}^{(i)}}$, guided by a specific pruning paradigm $C$ and sparsity $s^m\%$:

$$\kappa^m(\mathcal{G}^{(i)}) = \{\mathcal{V}^{(i)}, \mathcal{E}^{(i)} \setminus \mathcal{E}_p^{(i)}\}, \quad \mathcal{E}_p^{(i)} = \text{TopK}\left(-C^m(\mathcal{E}), \lceil |\mathcal{E}^{(i)}| \times s^m\% \rceil\right), \tag{8}$$

where $C^m(\cdot)$ acts as the $m$-th expert's scoring function that evaluates edge importance. We leverage long-tail gradient estimation (Liu et al., 2020) to ensure the $\text{TopK}(\cdot)$ operator is differentiable. Furthermore, to ensure different sparsity criteria drive the sparsifier, we implement $C^m(\cdot)$ as follows:

$$C^m(e_{ij}) = \text{FFN}\left(x_i, x_j, c(e_{ij})\right), \quad c^m(e_{ij}) \in \left\{ \begin{array}{c} \text{Degree: } \left(|\mathcal{N}(v_i) + \mathcal{N}(v_j)|\right)/2 \\ \text{Jaccard Similarity: } \frac{|\mathcal{N}(v_i) \cap \mathcal{N}(v_j)|}{|\mathcal{N}(v_i) \cup \mathcal{N}(v_j)|} \\ \text{ER: } (e_i - e_j)^T \mathbf{L}^{-1}(e_i - e_j) \\ \text{Gradient Magnitude: } |\partial\mathcal{L}/\partial e_{ij}| \end{array} \right\}, \tag{9}$$

where $\text{FFN}(\cdot)$ is a feed-forward network, $c^m(e_{ij})$ represents the prior guidance on edge significance. By equipping different sparsifiers with various priors and sparsity levels, we can customize the most appropriate pruning strategy for each node's local scenario. In practice, we select four widely-used pruning criteria including edge degree (Seo et al., 2024), Jaccard similarity (Murphy, 1996a; Satuluri et al., 2011b), effective resistance (Spielman & Srivastava, 2008; Liu et al., 2023b) and gradient magnitude (Wan & Schweitzer, 2021; Zhang et al., 2024a). Details regarding these criteria and their implementations are in Appendix B.

### 3.4 Graph Mixture on Grassmann Manifold

After employing $k$ sparsifiers driven by different criteria and sparsity levels, we are in need of an effective mechanism to ensemble these $k$ sparse subgraphs and maximize the aggregation of their advantages. A straightforward approach is voting or averaging (Sagi & Rokach, 2018); however, such simple merging may fail to capture the intricate relationships among multi-view graphs (Kang et al., 2020), potentially resulting in the loss of advantageous properties from all experts. Inspired by recent advances in manifold representations (Dong et al., 2013; Bendokat et al., 2024), we develop a subspace-based sparse graph ensembling mechanism. We first provide the definition of the Grassmann manifold (Bendokat et al., 2024) as follows:

**Definition 1** (Grassmann manifold). *Grassmann manifold $Gr(n, p)$ is the space of $n$-by-$p$ matrices (e.g., $\mathbf{M}$) with orthonormal columns, where $0 \leq p \leq n$, i.e.,*

$$Gr(n, p) = \left\{ \mathbf{M} | \mathbf{M} \in \mathbb{R}^{n \times p}, \mathbf{M}^\top \mathbf{M} = \mathbf{I} \right\}. \tag{10}$$

According to Grassmann manifold theory, each orthonormal matrix represents a unique subspace and thus corresponds to a distinct point on the Grassmann manifold (Lin et al., 2020). This applies to the eigenvector matrix of the normalized Laplacian matrix ($\mathbf{U} = \mathbf{L}[:, : p] \in \mathbb{R}^{n \times p}$), which comprises the first $p$ eigenvectors and is orthonormal (Merris, 1995), and thereby can be mapped onto the Grassmann manifold.

Consider the $k$ sparse subgraphs $\{\widetilde{\mathcal{G}}_m^{(i)}\}_{m=1}^k$, their subspace representations are $\{\mathbf{U}_m^{(i)} \in \mathbb{R}^{|\mathcal{N}(v_i)| \times p}\}_{m=1}^k$. We aim to identify an oracle subspace $\mathbf{U}^{(i)}$ on the Grassmann manifold, which essentially represents a graph, that serves as an informative combination of $k$ base graphs. Formally, we present the following objective function:

$$\min_{\mathbf{U}^{(i)} \in \mathbb{R}^{|\mathcal{N}(v_i)| \times p}} \sum_{m=1}^k \left( \underbrace{\operatorname{tr}(\mathbf{U^{(i)}}^\top \mathbf{L}_m \mathbf{U}^{(i)})}_{\text{(1) node connectivity}} + \underbrace{\overbrace{E_m^{(i)}}^{\text{expert score}} \cdot d^2(\mathbf{U}^{(i)}, \mathbf{U}_m^{(i)})}_{\text{(2) subspace distance}} \right), \text{s. t. } \mathbf{U^{(i)}}^\top \mathbf{U}^{(i)} = \mathbf{I} \tag{11}$$

where $\operatorname{tr}(\cdot)$ calculates the trace of matrices, $\mathbf{L}_m$ is the graph Laplacian of $\mathcal{G}_m^{(i)}$, $d^2(\mathbf{U}_1, \mathbf{U}_2)$ denotes the project distance between two subspaces (Dong et al., 2013), and $E_m^{(i)}$ is the expert score for the $m$-th expert, calculated by the routing network $\Psi$, which determines which expert's subspace the combined subspace should more closely align with. In Equation (11), the first term is designed to preserve the original node connectivity based on spectral embedding, and the second term controls that individual subspaces are close to the final representative subspace $\mathbf{U}^{(i)}$. Using the Rayleigh-Ritz Theorem (Jia & Stewart, 2001), we provide a closed-form solution for Equation (11) and obtain the graph Laplacian of the ensemble sparse graph $\widehat{\mathcal{G}^{(i)}}$ as follows:

$$\widehat{\mathbf{L}^{(i)}} = \sum_{m=1}^k \left( \mathbf{L}_m - E_m^{(i)} \cdot \mathbf{U^{(i)}}^\top \mathbf{U}^{(i)} \right). \tag{12}$$

We provide detailed derivations and explanations for Equations (11) and (12) in Appendix C. Consequently, we can reformulate the function $\texttt{ESMB}(\cdot)$ in Equation (5) as follows:

$$\texttt{ESMB}(\{\widetilde{\mathcal{G}}_m^{(i)}\}_{m=1}^k) = \{\mathbf{D}^{(i)} - \widehat{\mathbf{L}^{(i)}}, \mathbf{X}^{(i)}\} = \left\{ \mathbf{D}^{(i)} - \sum_{m=1}^k \left( \mathbf{L}_m - E_m^{(i)} \cdot \mathbf{U^{(i)}}^\top \mathbf{U}^{(i)} \right), \mathbf{X}^{(i)} \right\}, \tag{13}$$

where $\mathbf{D}^{(i)}$ is the degree matrix of $v_i$'s ego-graph. On the Grassmann manifold, the subspace ensemble effectively captures the beneficial properties of each expert's sparse graph. After obtaining the final version of each node's ego-graph, $\widehat{\mathcal{G}^{(i)}} = \{\widehat{\mathbf{A}^{(i)}}, \mathbf{X}^{(i)}\}$, we conduct a post-sparsification step as the graph ensembled on the Grassmann manifold can become dense again. Specifically, we obtain the final sparsity $s^{(i)}\%$ for $v_i$ by weighting the sparsity of each expert and sparsifying $\widehat{\mathcal{G}^{(i)}}$.

$$\widehat{\mathcal{G}^{(i)}} \leftarrow \{\text{TopK}(\widehat{\mathbf{A}^{(i)}}, |\mathcal{E}^{(i)}| \times s^{(i)}\%), \mathbf{X}^{(i)}\}, \ s^{(i)}\% = \frac{1}{k} \sum_{m=1}^k s^m\%. \tag{14}$$

Post-sparsified $\widehat{\mathcal{G}^{(i)}}$ are then reassembled together into $\widehat{\mathcal{G}} \leftarrow \{\widehat{\mathcal{G}^{(1)}}, \widehat{\mathcal{G}^{(2)}}, \cdots, \widehat{\mathcal{G}^{(|\mathcal{V}|)}}\}$, with detailed explanations in appendix I. Ultimately, the sparsified graph $\widehat{\mathcal{G}}$ produced by MoG can be input into any MPNN (Gilmer et al., 2017) or graph transformer (Min et al., 2022) for end-to-end training.

### 3.5 TRAINING AND OPTIMIZATION

**Additional Loss Functions** Following classic MoE works (Shazeer et al., 2017; Wang et al., 2024), we introduce an expert importance loss to prevent MoG from converging to a trivial solution where only a single group of experts is consistently selected:

$$\text{Importance}(\mathcal{V}) = \sum_{i=1}^{|\mathcal{V}|} \sum_{j=1}^{k} E_m^{(i)}, \quad \mathcal{L}_{\text{importance}}(\mathcal{V}) = \text{CV}(\text{Importance}(\mathcal{V}))^2, \quad (15)$$

where $\text{Importance}(\mathcal{V})$ represents the sum of each node's expert scores across the node-set, $\text{CV}(\cdot)$ calculates the coefficient of variation, and $\mathcal{L}_{\text{importance}}$ ensures the variation of experts. Therefore, the final loss function combines both task-specific and MoG-related losses, formulated as follows:

$$\mathcal{L} = \mathcal{L}_{\text{task}} + \lambda \cdot \mathcal{L}_{\text{importance}}, \quad (16)$$

where $\lambda$ is a hand-tuned scaling factor, with its sensitivity analysis placed in Section 4.4.

**Complexity Analysis** To better illustrate the effectiveness and clarity of MoG, we provide a comprehensive algorithmic table in Appendix D and detailed complexity analysis in Appendix E. To address concerns regarding the runtime efficiency of MoG, we have included an empirical analysis of efficiency in Section 4.5.

## 4 EXPERIMENTS

In this section, we conduct extensive experiments to answer the following research questions: (**RQ1**) Can MoG effectively help GNNs combat graph sparsity? (**RQ2**) Does MoG genuinely accelerate the GNN inference? (**RQ3**) Can MoG help boost GNN performance? (**RQ4**) How sensitive is MoG to its key components and parameters?

### 4.1 EXPERIMENT SETUP

**Datasets and Backbones** We opt for four large-scale OGB benchmarks (Hu et al., 2020), including OGBN-ARXIV, OGBN-PROTEINS and OGBN-PRODUCTS for node classification, and OGBG-PPA for graph classification. The dataset splits are given by (Hu et al., 2020). Additionally, we choose two superpixel datasets, MNIST and CIFAR-10 (Knyazev et al., 2019). We select GraphSAGE (Hamilton et al., 2017), DeeperGCN (Li et al., 2020a), and PNA (Corso et al., 2020) as the GNN backbones. More details are provided in Appendix F.

**Parameter Configurations** For MoG, we adopt the $m = 4$ sparsity criteria outlined in Section 3.3, assigning $n = 3$ different sparsity levels $\{s_1, s_2, s_3\}$ to each criterion, resulting in a total of $K = m \times n = 12$ experts. We select $k = 2$ sparsifier experts for each node, and set the loss scaling factor $\lambda = 1e - 2$ across all datasets and backbones. By adjusting the sparsity combination, we can control the global sparsity of the entire graph. We present more details on parameter settings in Appendix F.4, and a recipe for adjusting the graph sparsity in Appendix F.6.

### 4.2 MOG AS GRAPH SPARSIFIER (RQ1 & RQ2)

To answer **RQ1** and **RQ2**, we comprehensively compare MoG with eleven widely-used topology-guided sparsifiers and five semantic-guided sparsifiers, as outlined in Table 1, with more detailed explanations in Appendix F.5. The quantitative results on five datasets are shown in Tables 1 and 9 to 12 and the efficiency comparison is in Figure 3. We give the following observations (**Obs.**):

**Obs. ❶ MoG demonstrates superior performance in both transductive and inductive settings.** As shown in Tables 1, 2 and 9 to 11, MoG outperforms other sparsifiers in both transductive and inductive settings. Specifically, for node classification tasks, MoG achieves a $0.09\%$ performance improvement while sparsifying $30\%$ of the edges on OGBN-PROTEINS+GraphSAGE. Even when sparsifying $50\%$ of the edges on OGBN-PROTEINS+DeeperGCN, the ROC-AUC only drops by $0.81\%$. For graph classification tasks, MoG can remove up to $50\%$ of the edges on MNIST with a $0.14\%$ performance improvement, surpassing other sparsifiers by $0.99\% \sim 12.97\%$ in accuracy.

**Obs. ❷ Different datasets and backbones exhibit varying sensitivities to sparsification.** As shown in Tables 1 and 10, despite OGBN-PROTEINS being relatively insensitive to sparsification, sparsification at extremely high levels (e.g., $70\%$) causes more performance loss for GraphSAGE

Table 1: Node classification performance comparison to state-of-the-art sparsification methods. All methods are trained using **GraphSAGE**, and the reported metrics represent the average of **five runs**. We denote methods with † that do not have precise control over sparsity; their performance is reported around the target sparsity $\pm 2\%$. "Sparsity %" refers to the ratio of removed edges as defined in Section 2. "OOM" and "OOT" denotes out-of-memory and out-of-time, respectively.

| | Dataset | OGBN-ARXIV (Accuracy↑) | | | | OGBN-PROTEINS (ROC-AUC↑) | | | |
|---|---|---|---|---|---|---|---|---|---|
| | Sparsity % | 10 | 30 | 50 | 70 | 10 | 30 | 50 | 70 |
| Topology-guided | Random | $70.03_{\downarrow 1.46}$ | $68.40_{\downarrow 3.09}$ | $64.32_{\downarrow 7.17}$ | $61.18_{\downarrow 10.3}$ | $76.72_{\downarrow 0.68}$ | $75.03_{\downarrow 2.37}$ | $73.58_{\downarrow 3.82}$ | $72.30_{\downarrow 5.10}$ |
| | Rank Degree† (Voudigari et al., 2016) | $68.13_{\downarrow 3.36}$ | $67.01_{\downarrow 4.48}$ | $65.58_{\downarrow 5.91}$ | $62.17_{\downarrow 9.32}$ | $77.47_{\uparrow 0.07}$ | $76.15_{\downarrow 1.25}$ | $75.59_{\downarrow 1.81}$ | $74.23_{\downarrow 3.17}$ |
| | Local Degree† (Hamann et al., 2016) | $68.94_{\downarrow 2.55}$ | $67.20_{\downarrow 4.29}$ | $65.45_{\downarrow 6.04}$ | $65.59_{\downarrow 5.90}$ | $76.20_{\downarrow 1.20}$ | $76.05_{\downarrow 1.35}$ | $76.09_{\downarrow 1.31}$ | $72.88_{\downarrow 4.52}$ |
| | Forest Fire† (Leskovec et al., 2006) | $68.39_{\downarrow 3.10}$ | $68.10_{\downarrow 3.39}$ | $67.36_{\downarrow 4.13}$ | $65.22_{\downarrow 6.27}$ | $76.50_{\downarrow 0.90}$ | $75.37_{\downarrow 2.03}$ | $74.29_{\downarrow 3.11}$ | $72.11_{\downarrow 5.29}$ |
| | G-Spar (Murphy, 1996b) | $71.30_{\downarrow 0.19}$ | $69.29_{\downarrow 2.20}$ | $65.56_{\downarrow 5.93}$ | $65.49_{\downarrow 6.00}$ | $77.38_{\downarrow 0.02}$ | $77.36_{\downarrow 0.04}$ | $76.02_{\downarrow 1.38}$ | $75.89_{\downarrow 1.51}$ |
| | LSim† (Satuluri et al., 2011a) | $69.22_{\downarrow 2.27}$ | $66.15_{\downarrow 5.34}$ | $61.07_{\downarrow 10.4}$ | $60.32_{\downarrow 11.2}$ | $76.83_{\downarrow 0.57}$ | $76.01_{\downarrow 1.39}$ | $74.83_{\downarrow 2.57}$ | $73.65_{\downarrow 3.75}$ |
| | SCAN (Xu et al., 2007) | $71.55_{\uparrow 0.06}$ | $69.27_{\downarrow 2.22}$ | $65.14_{\downarrow 6.35}$ | $64.72_{\downarrow 6.77}$ | $77.60_{\downarrow 0.20}$ | $76.88_{\downarrow 0.52}$ | $76.19_{\downarrow 1.21}$ | $74.32_{\downarrow 3.08}$ |
| | ER (Spielman & Srivastava, 2008) | $71.63_{\uparrow 0.14}$ | $69.48_{\downarrow 2.01}$ | $69.00_{\downarrow 2.49}$ | $67.15_{\downarrow 4.34}$ | | | OOT | |
| | DSpar (Liu et al., 2023b) | $71.23_{\downarrow 0.26}$ | $68.50_{\downarrow 2.99}$ | $64.79_{\downarrow 6.70}$ | $63.11_{\downarrow 8.38}$ | $77.34_{\downarrow 0.06}$ | $77.06_{\downarrow 0.34}$ | $76.38_{\downarrow 1.02}$ | $75.49_{\downarrow 1.91}$ |
| Semantic-guided | UGS† (Chen et al., 2021) | $68.77_{\downarrow 2.72}$ | $66.30_{\downarrow 5.19}$ | $65.72_{\downarrow 5.77}$ | $63.10_{\downarrow 8.39}$ | $76.80_{\downarrow 0.60}$ | $75.46_{\downarrow 1.94}$ | $73.28_{\downarrow 4.12}$ | $73.31_{\downarrow 4.09}$ |
| | GEBT (You et al., 2022) | $69.04_{\downarrow 2.45}$ | $65.29_{\downarrow 6.20}$ | $65.88_{\downarrow 5.61}$ | $65.62_{\downarrow 5.87}$ | $76.30_{\downarrow 1.10}$ | $76.17_{\downarrow 1.23}$ | $74.43_{\downarrow 2.97}$ | $74.12_{\downarrow 3.28}$ |
| | MGSpar (Wan & Schweitzer, 2021) | $70.22_{\downarrow 1.27}$ | $69.13_{\downarrow 2.36}$ | $68.27_{\downarrow 3.22}$ | $66.55_{\downarrow 4.94}$ | | | OOM | |
| | ACE-GLT† (Wang et al., 2023b) | $71.88_{\uparrow 0.39}$ | $70.14_{\downarrow 1.35}$ | $68.08_{\downarrow 3.41}$ | $67.04_{\downarrow 4.45}$ | $77.59_{\downarrow 0.19}$ | $76.14_{\downarrow 1.26}$ | $75.43_{\downarrow 1.97}$ | $73.28_{\downarrow 4.12}$ |
| | WD-GLT† (Hui et al., 2023) | $71.92_{\uparrow 0.43}$ | $70.21_{\downarrow 1.28}$ | $68.30_{\downarrow 3.19}$ | $66.57_{\downarrow 4.92}$ | | | OOM | |
| | AdaGLT (Zhang et al., 2024a) | $71.22_{\downarrow 0.27}$ | $70.18_{\downarrow 1.31}$ | $\mathbf{69.13}_{\downarrow 2.36}$ | $67.02_{\downarrow 4.47}$ | $77.49_{\downarrow 0.09}$ | $76.76_{\downarrow 1.64}$ | $76.00_{\downarrow 2.40}$ | $75.44_{\downarrow 2.96}$ |
| | **MoG (Ours)†** | $\mathbf{71.93}_{\uparrow 0.44}$ | $\mathbf{70.53}_{\downarrow 0.96}$ | $69.06_{\downarrow 2.43}$ | $\mathbf{67.31}_{\downarrow 4.18}$ | $\mathbf{77.78}_{\uparrow 0.38}$ | $\mathbf{77.49}_{\uparrow 0.09}$ | $\mathbf{76.46}_{\downarrow 0.94}$ | $\mathbf{76.12}_{\downarrow 1.28}$ |
| | Whole Dataset | | $71.49_{\pm 0.01}$ | | | | $77.40_{\pm 0.1}$ | | |

Table 2: Graph classification performance comparison to state-of-the-art sparsification methods. The reported metrics represent the average of **five runs**.

| | Dataset | MNIST + PNA (Accuracy ↑) | | | | OGBN-PPA + DeeperGCN (Accuracy ↑) | | | |
|---|---|---|---|---|---|---|---|---|---|
| | Sparsity % | 10 | 30 | 50 | 70 | 10 | 30 | 50 | 70 |
| Topology-guided | Random | $94.61_{\downarrow 2.74}$ | $87.23_{\downarrow 10.1}$ | $84.82_{\downarrow 12.5}$ | $80.07_{\downarrow 17.3}$ | $75.44_{\downarrow 1.65}$ | $73.81_{\downarrow 4.09}$ | $71.97_{\downarrow 5.12}$ | $69.62_{\downarrow 7.47}$ |
| | Rank Degree† (Voudigari et al., 2016) | $96.42_{\downarrow 0.93}$ | $94.23_{\downarrow 3.12}$ | $92.36_{\downarrow 4.99}$ | $89.20_{\downarrow 8.15}$ | $75.81_{\downarrow 1.28}$ | $74.99_{\downarrow 2.10}$ | $74.12_{\downarrow 2.97}$ | $70.68_{\downarrow 6.41}$ |
| | Local Degree† (Hamann et al., 2016) | $95.95_{\downarrow 1.40}$ | $93.37_{\downarrow 3.98}$ | $90.11_{\downarrow 7.24}$ | $86.24_{\downarrow 11.1}$ | $76.43_{\downarrow 0.66}$ | $75.87_{\downarrow 1.22}$ | $72.11_{\downarrow 4.98}$ | $69.93_{\downarrow 7.16}$ |
| | Forest Fire† (Leskovec et al., 2006) | $96.75_{\downarrow 0.60}$ | $95.42_{\downarrow 1.93}$ | $95.03_{\downarrow 2.32}$ | $93.10_{\downarrow 4.25}$ | $76.38_{\downarrow 0.71}$ | $75.33_{\downarrow 1.76}$ | $73.18_{\downarrow 3.91}$ | $71.49_{\downarrow 5.60}$ |
| | G-Spar (Murphy, 1996b) | $97.10_{\downarrow 0.25}$ | $96.59_{\downarrow 0.76}$ | $94.36_{\downarrow 2.99}$ | $92.48_{\downarrow 4.87}$ | $77.68_{\uparrow 0.59}$ | $73.90_{\downarrow 3.19}$ | $69.52_{\downarrow 7.57}$ | $68.10_{\downarrow 8.99}$ |
| | LSim† (Satuluri et al., 2011a) | $95.79_{\downarrow 1.56}$ | $92.14_{\downarrow 5.21}$ | $92.29_{\downarrow 5.06}$ | $91.95_{\downarrow 5.40}$ | $76.04_{\downarrow 1.05}$ | $74.40_{\downarrow 2.69}$ | $72.78_{\downarrow 4.31}$ | $68.21_{\downarrow 8.88}$ |
| | SCAN (Xu et al., 2007) | $95.81_{\downarrow 1.54}$ | $93.48_{\downarrow 3.87}$ | $90.18_{\downarrow 7.17}$ | $86.48_{\downarrow 10.9}$ | $75.23_{\downarrow 1.86}$ | $75.18_{\downarrow 1.91}$ | $72.48_{\downarrow 4.61}$ | $71.11_{\downarrow 5.98}$ |
| | ER (Spielman & Srivastava, 2008) | $94.77_{\downarrow 2.58}$ | $93.91_{\downarrow 3.44}$ | $93.45_{\downarrow 3.90}$ | $91.07_{\downarrow 6.28}$ | $77.94_{\uparrow 0.85}$ | $75.15_{\downarrow 1.94}$ | $73.23_{\downarrow 3.86}$ | $72.74_{\downarrow 4.35}$ |
| | DSpar (Liu et al., 2023b) | $94.97_{\downarrow 2.38}$ | $93.80_{\downarrow 3.55}$ | $92.23_{\downarrow 5.12}$ | $90.48_{\downarrow 6.87}$ | $76.33_{\downarrow 0.94}$ | $73.37_{\downarrow 3.72}$ | $72.98_{\downarrow 4.11}$ | $70.77_{\downarrow 6.32}$ |
| Semantic | ICPG (Sui et al., 2023) | $97.69_{\uparrow 0.34}$ | $97.39_{\uparrow 0.04}$ | $96.80_{\downarrow 0.55}$ | $93.77_{\downarrow 3.58}$ | $77.36_{\uparrow 0.27}$ | $75.24_{\downarrow 1.85}$ | $73.18_{\downarrow 3.91}$ | $71.09_{\downarrow 6.00}$ |
| | AdaGLT (Zhang et al., 2024a) | $97.31_{\downarrow 0.04}$ | $96.58_{\downarrow 0.77}$ | $94.14_{\downarrow 3.21}$ | $92.08_{\downarrow 5.27}$ | $76.22_{\downarrow 0.87}$ | $73.54_{\downarrow 3.55}$ | $70.10_{\downarrow 6.99}$ | $69.28_{\downarrow 7.81}$ |
| | **MoG (Ours)†** | $\mathbf{97.80}_{\uparrow 0.45}$ | $\mathbf{97.74}_{\uparrow 0.39}$ | $\mathbf{97.79}_{\uparrow 0.44}$ | $\mathbf{95.30}_{\downarrow 2.05}$ | $\mathbf{78.43}_{\downarrow 1.34}$ | $\mathbf{77.90}_{\uparrow 0.81}$ | $\mathbf{75.23}_{\downarrow 1.86}$ | $\mathbf{73.09}_{\downarrow 4.00}$ |
| | Whole Dataset | | $97.35_{\pm 0.07}$ | | | | $77.09_{\pm 0.04}$ | | |

compared to DeeperGCN, with the former experiencing a $2.28\%$ drop and the latter only $1.07\%$, which demonstrates the varying sensitivity of different GNN backbones to sparsification. Similarly, we observe in Table 2 that the MNIST dataset shows a slight accuracy increase even with $50\%$ sparsification, whereas the OGBG-PPA dataset suffers a $1.86\%$ performance decline, illustrating the different sensitivities to sparsification across graph datasets.

**Obs. ❸ MoG can effectively accelerate GNN inference with negligible performance loss.** Figure 3 illustrates the actual acceleration effects of MoG compared to other baseline sparsifiers. It is evident that MoG achieves $1.6\times$ *lossless acceleration* on OGBN-PROTEINS+DeeperGCN and OGBN-PRODUCTS+GraphSAGE, meaning the performance is equal to or better than the vanilla backbone. Notably, on OGBN-PRODUCTS+DeeperGCN, MoG achieves $3.3\times$ acceleration with less than a $1.0\%$ performance drop. Overall, MoG provides significantly superior inference acceleration compared to its competitors.

### 4.3 MoG AS PERFORMANCE BOOSTER (RQ3)

In the context of **RQ3**, MoG is developed to augment GNN performance by selectively removing a limited amount of noisy and detrimental edges, while simultaneously preventing excessive sparsification that could degrade GNN performance. Consequently, we uniformly set the sparsity combination to $\{90\%, 85\%, 80\%\}$. We combine MoG with state-of-the-art GNNs on both node-level

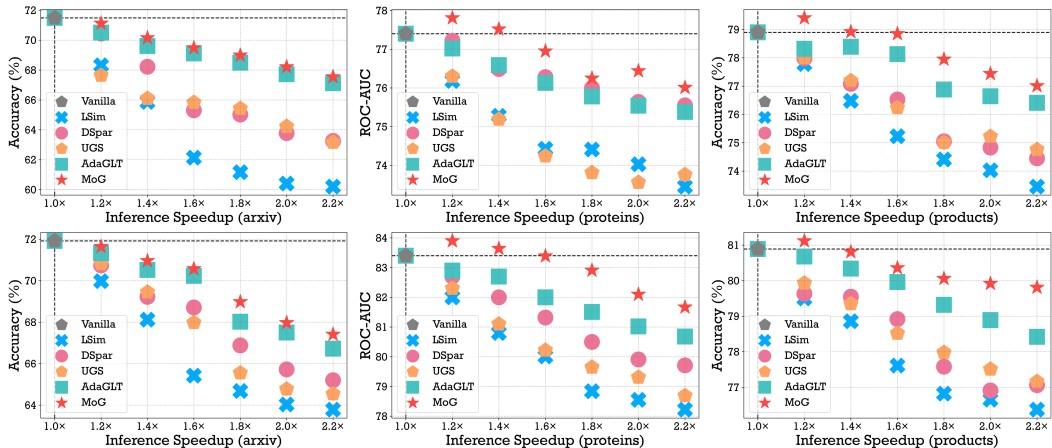

Figure 3: The trade-off between inference speedup and model performance for MoG and other sparsifiers. The first and second rows represent results on GraphSAGE and DeeperGCN, respectively. The gray pentagon represents the performance of the original GNN without sparsification.

Table 3: Node classification results on OGBN-PROTEINS with RevGNN and GAT+BoT and graph classification results on OGBG-PPA with PAS and DeeperGCN. Mean and standard deviation values from **five** random runs are presented.

| Model | OGBN-PROTEINS (ROC-AUC↑) | | OGBG-PPA (Accuracy↑) | |
|---|---|---|---|---|
| | RevGNN | GAT+BoT | PAS | DeeperGCN |
| w/o MoG | 88.14± 0.24 | 88.09 ± 0.16 | 78.28 ± 0.24 | 77.09 ± 0.04 |
| w/ MoG | **89.04** ± 0.72 (Sparsity: 9.2%) | **88.72** ± 0.50 (Sparsity: 12.7%) | **78.66** ± 0.47 (Sparsity: 6.6%) | **78.43** ± 0.19 (Sparsity: 10.8%) |

and graph-level tasks. The former include RevGNN (Li et al., 2021) and GAT+BoT (Wang et al., 2021), which rank fourth and seventh, respectively, on the OGBN-PROTEINS benchmark, and the latter include PAS (Wei et al., 2021) and DeeperGCN (Li et al., 2020a), ranking fourth and sixth on the OGBN-PPA benchmark. We observe from Table 3:

**Obs. ❹ MoG can assist the "top-student" backbones to learn better.** Despite RevGNN and PAS being high-ranking backbones for OGBN-PROTEINS and OGBG-PPA, MoG still achieves non-marginal performance improvements through moderate graph sparsification: $1.02\%\uparrow$ on RevGNN+OGBN-PROTEINS and $1.74\%\uparrow$ on DeeperGCN+OGBG-PPA. This demonstrates that MoG can effectively serve as a plugin to boost GNN performance by setting a relatively low sparsification rate.

## 4.4 SENSITIVITY ANALYSIS (RQ4)

To answer **RQ4**, we perform a sensitivity analysis on the two most important parameters in MoG: the number of selected experts $k$ and the expert importance loss coefficient $\lambda$. We compared the performance of MoG when choosing different numbers of experts per node, as outlined in Figure 4. The effect of different scaling factors $\lambda$ on OGBN-PROTEINS+DeeperGCN is shown in Table 4. Based on the results of the above sensitivity analysis, we observe that:

**Obs. ❺ Sparse expert selection helps customized sparsification.** It can be observed FROM Figure 4 that the optimal $k$ varies with the level of graph sparsity. At lower sparsity (10%), $k = 1$ yields relatively good performance. However, as sparsity increases to 50%, model performance peaks at $k = 4$, suggesting that in high sparsity environments, more expert opinions contribute to better sparsification. Notably, when $k$ increases to 6, MoG's performance declines, indicating that a more selective approach in sparse expert selection aids in better model generalization. For a balanced consideration of performance and computational efficiency, we set $k = 2$ in all experiments. We further provide sensitivity analysis results of parameter $k$ on more datasets, as shown in Appendix G.2.

**Obs. ❻ Sparsifier load balancing is essential.** We conduct a sensitivity analysis of the expert importance loss coefficient $\lambda$. A larger $\lambda$ indicates greater variation in the selected experts. As shown

in Table 4, $\lambda = 0$ consistently resulted in the lowest performance, as failing to explicitly enforce variation among experts leads to the model converging to a trivial solution with the same set of experts (Wang et al., 2024; Shazeer et al., 2017). Conversely, $\lambda = 1e - 1$ performed slightly better than $\lambda = 1e - 2$ at higher sparsity levels, supporting the findings in Obs. 5 that higher sparsity requires more diverse sparsifier experts.

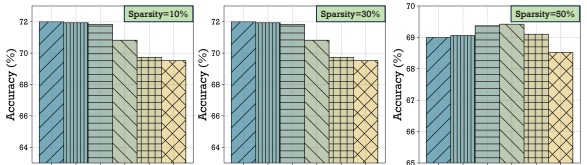

| $\lambda$ | 0 | 1e-2 | 1e-1 |
|---|---|---|---|
| 10% | $81.19_{\pm 0.08}$ | $\mathbf{83.32}_{\pm 0.19}$ | $83.04_{\pm 0.23}$ |
| 30% | $79.77_{\pm 0.08}$ | $\mathbf{82.14}_{\pm 0.23}$ | $82.08_{\pm 0.25}$ |
| 50% | $79.40_{\pm 0.06}$ | $81.79_{\pm 0.21}$ | $\mathbf{82.04}_{\pm 0.20}$ |
| 70% | $78.22_{\pm 0.13}$ | $80.90_{\pm 0.24}$ | $\mathbf{80.97}_{\pm 0.28}$ |

Figure 4: Sensitivity study on parameter $k$, *i.e.*, how many experts are chosen per node. The results are reported based on OGBN-ARXIV+GraphSAGE.

Table 4: Sensitivity study on scaling factor $\lambda$. The results are reported on OGBN-PROTEINS+DeeperGCN.

Table 5: Running time efficiency comparison on OGBN-PRODUCTS+GraphSAGE. We consistently set $n = 4, k = 2$, corresponding to utilizing 4 pruning criteria and selecting 2 experts for each node, and vary $m \in \{1, 2, 3\}$ to check how the training cost grows with $m$ increasing.

| Sparsity | 30% | | 50% | |
|---|---|---|---|---|
| Metric | Per-epoch Time (s) | Accuracy (%) | Per-epoch Time (s) | Accuracy (%) |
| Random | $18.71 \pm 0.14$ | $74.21 \pm 0.28$ | $15.42 \pm 0.24$ | $71.08 \pm 0.34$ |
| AdaGLT | $23.55 \pm 0.20$ | $77.30 \pm 0.54$ | $21.68 \pm 0.26$ | $74.38 \pm 0.79$ |
| MoG($m = 1, K = 3$) | $20.18 \pm 0.14$ | $77.75 \pm 0.22$ | $18.19 \pm 0.30$ | $76.10 \pm 0.49$ |
| MoG($m = 2, K = 6$) | $21.25 \pm 0.22$ | $78.23 \pm 0.29$ | $19.70 \pm 0.30$ | $76.43 \pm 0.49$ |
| MoG($m = 3, K = 12$) | $23.19 \pm 0.18$ | $78.15 \pm 0.32$ | $20.83 \pm 0.29$ | $76.98 \pm 0.49$ |

### 4.5 EFFICIENCY ANALYSIS AND ABLATION STUDY (RQ4)

**Efficiency Analysis** To verify that MoG can achieve better results with less additional training cost than the previous SOTA methods, we compare the accuracy and the time efficiency of MoG with AdaGLT on OGBN-PRODUCT+GraphSAGE, as outlined in Table 5. We have:

**Obs. ❼ MoG can achieve better accuracy with less additional training cost.** It is evident in Table 5 that MoG incurs less additional training cost compared to AdaGLT while achieving significant improvements in sparsification performance. More importantly, we demonstrate that with $k = 2$, MoG does not incur significantly heavier training burdens as the number of sparsifiers increases. Specifically, at $s\% = 50\%$, the difference in per epoch time between MoG ($K = 3$) and MoG ($K = 12$) is only 2.63 seconds, consistent with the findings of mainstream sparse MoE approaches (Wang et al., 2024).

**Ablation Study** We test three different settings of $\epsilon$ (in Equation (3)) on OGBN-ARXIV+GraphSAGE: (1) $\epsilon \sim \mathcal{N}(0, \mathbf{I})$, (2) $\epsilon = 0$, and (3) $\epsilon = 0.2$, presented in Table 15. Our key finding is that randomness in gating networks consistently benefits our model. More results and detailed analysis can be found in Appendix G.3.

## 5 CONCLUSION & LIMITATION

In this paper, we introduce a new graph sparsification paradigm termed MoG, which leverages multiple graph sparsifiers, each equipped with distinct sparsity levels and pruning criteria. MoG selects the most suitable sparsifier expert based on each node's local context, providing a customized graph sparsification solution, followed by an effective mixture mechanism on the Grassmann manifold to ensemble the sparse graphs produced by various experts. Extensive experiments on four large-scale OGB datasets and two superpixel datasets have rigorously demonstrated the effectiveness of MoG. A potential limitation of MoG is its current reliance on 1-hop decomposition to represent each node's local context. The performance of extending this approach to $k$-hop contexts remains unexplored, suggesting a possible direction for future research.

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

## A    NOTATIONS

We conclude the commonly used notations throughout the manuscript in Table 6.

## B    DEATILS ON PRUNING CRITERIA

In this section, we will thoroughly explain the four pruning criteria we selected and the rationale behind these choices.

Table 6: The notations that are commonly used in the manuscript.

| Notation | Definition |
|---|---|
| $\mathcal{G} = \{\mathcal{V}, \mathcal{E}\} = \{\mathbf{A}, \mathbf{X}\}$ | Input graph |
| $\mathbf{A}$ | Input adjacency matrix |
| $\mathbf{X}$ | Node feature matrix |
| $\mathbf{L}$ | Graph Laplacian matrix |
| $\text{COMB}(\cdot)$ | GNN ego-node transformation function |
| $\text{AGGR}(\cdot)$ | GNN message aggregation function |
| $\text{ESM}(\cdot)$ | Sparse graph combination on function |
| $s\%$ | Sparsity ratio (the ratio of removed edges) |
| $v_i$ | The $i$-th node in $\mathcal{G}$ |
| $x_i$ | Node feature vector for $v_i$ |
| $\mathcal{G}^{(i)}$ | The 1-hop ego-graph for $v_i$ |
| $\phi(\mathcal{G}^{(i)})$ | Routing network |
| $K$ | The number of total sparsifier experts |
| $k$ | The number of selected sparsifier experts per node |
| $W_g, W_n$ | Trainable parameters in the routing network |
| $\kappa(\mathcal{G})$ | A graph sparsifier |
| $\mathcal{G}_m^{(i)}$ | The sparse ego-graph of $v_i$ produced by the $m$-th graph sparsifier |
| $\widehat{\mathcal{G}^{(i)}} = \{\widehat{\mathcal{V}^{(i)}}, \widehat{\mathcal{E}^{(i)}}\}$ | The ensemble sparse graph produced by MoG for $v_i$ |
| $\mathcal{E}_p^{(i)}$ | Edges removed surrounding $v_i$ |
| $c^m(e_{ij})$ | Prior guidance on edge importance $e_{ij}$ |

- **Edge degree** of $e_{ij}$ is defined as follows:

$$\text{Degree}\,(e_{ij}) = \frac{1}{2}\left(|\mathcal{N}(v_i) + \mathcal{N}(v_j)|\right). \tag{17}$$

Previous methods (Wang et al., 2022b; Seo et al., 2024) have explicitly or implicitly used edge degree for graph sparsification. Intuitively, edges with higher degrees are more replaceable. (Wang et al., 2022b) further formalizes this intuition from the perspective of bridge edges.

- **Jaccard Similarity** (Murphy, 1996b) measures the similarity between two sets by computing the portion of shared neighbors between two nodes ($v_i$ and $v_j$), as defined below:

$$\text{JaccradSimilarity}\,(v_i, v_j) = \frac{|\mathcal{N}(v_i) \cap \mathcal{N}(v_j)|}{|\mathcal{N}(v_i) \cup \mathcal{N}(v_j)|}. \tag{18}$$

Jaccard similarity is widely used for its capacity for detecting clusters, hubs, and outliers on social networks (Murphy, 1996b; Xu et al., 2007; Satuluri et al., 2011a).

- **Effective Resistance**, derived from the analogy to electrical circuits, is applied to graphs where edges represent resistors. The effective resistance of an edge is defined as the potential difference generated when a unit current is introduced at one vertex and withdrawn from the other. Once the effective resistance is calculated, a sparsified subgraph can be constructed by selecting edges with probabilities proportional to their effective resistances. Notably, (Spielman & Srivastava, 2008) proved that the quadratic form of the Laplacian for such sparsified graphs closely approximates that of the original graph. Consequently, the following inequality holds for the sparsified subgraph with high probability:

$$\forall \boldsymbol{x} \in \mathbb{R}^{|\mathcal{V}|} \quad (1-\epsilon)\boldsymbol{x}^T\mathbf{L}\boldsymbol{x} \le \boldsymbol{x}^T\tilde{\mathbf{L}}\boldsymbol{x} \le (1+\epsilon)\boldsymbol{x}^T\mathbf{L}\boldsymbol{x}, \tag{19}$$

where $\tilde{\mathbf{L}}$ is the Laplacian of the sparsified graph, and $\epsilon > 0$ is a small number. The insight is that effective resistance reflects the significance of an edge. Effective resistance aims to preserve the quadratic form of the graph Laplacian. This property makes it suitable for applications relying on

the quadratic form of the graph Laplacian, such as min-cut/max-flow problems. For computation simplicity, we do not directly utilize the definition of effective resistance, and use its approximation version (Liu et al., 2023b).

- **Gradient Magnitude**, a widely used pruning criterion, is prevalent not only in the field of graph sparsification but also in classical neural network pruning. Numerous studies (Lee et al., 2018; Tessera et al., 2021; Dettmers et al., 2019) leverage gradient magnitude to estimate parameter importance. Specifically for graph sparsification, MGSpar (Wan & Schweitzer, 2021) was the first to propose using meta-gradient to estimate edge importance. We consider gradient magnitude a crucial indicator of the graph's topological structure during training. Therefore, we explicitly design some sparsifier experts to focus on this information.

## C  GRAPH MIXTURE ON GRASSMANN MANIFOLD

In this section, we detail how we leverage the concept of Grassmann Manifold to effectively combine different sparse (ego-)graphs output by various sparsifiers.

According to Equation (10), each orthonormal matrix represents a unique subspace and thus corresponds to a distinct point on the Grassmann manifold (Lin et al., 2020). This applies to the eigenvector matrix of the normalized Laplacian matrix ($\mathbf{U} = \mathbf{L}[:,:p] \in \mathbb{R}^{n \times p}$), which comprises the first $p$ eigenvectors and is orthonormal (Merris, 1995), and thereby can be mapped onto the Grassmann manifold. Additionally, each row of the eigenvector matrix encapsulates the spectral embedding of each node in a p-dimensional space, where adjacent nodes have similar embedding vectors. This subspace representation, summarizing graph information, is applicable to various tasks such as clustering, classification, and graph merging (Dong et al., 2013).

In the context of MoG, we aim to efficiently find the final version that aggregates all the excellent properties of each point's $k$ versions of sparse ego-graph $\{\widetilde{\mathcal{G}}_m^{(i)}\}_{m=1}^{k}$ on the Grassmann Manifold. Moreover, this should guided by the expert scores computed by the routing network in Section 3.2. Let $\mathbf{D}_m$ and $\mathbf{A}_m$ denote the degree matrix and the adjacency matrix for $\widetilde{\mathcal{G}}_m^{(i)}$ (we omit the superscript $(\cdot)^{(i)}$ denoting $v_i$ for simplicity in the subsequent expressions), then the normalized graph Laplacian is defined as:

$$\mathbf{L}_m = \mathbf{D}_m^{-\frac{1}{2}}(\mathbf{D}_m - \mathbf{A}_m)\mathbf{D}_m^{\frac{1}{2}}. \tag{20}$$

Given the graph Laplacian $\mathbf{L}_m$ for each sparse graph, we calculate the spectral embedding matrix $\mathbf{U}_m$ through trace minimization:

$$\min_{\mathbf{U}_m \in \mathbb{R}^{|\mathcal{N}(v_m)| \times p}} \operatorname{tr}(\mathbf{U}_m^\top \mathbf{L}_m \mathbf{U}_m), \ \text{ s.t. } \mathbf{U}_m^\top \mathbf{U}_m = \mathbf{I}, \tag{21}$$

which can be solved by the Rayleigh-Ritz theorem. As mentioned above, each point on the Grassmann manifold can be represented by an orthonormal matrix $\mathbf{Y} \in \mathbb{R}^{|\mathcal{N}(v_i)| \times p}$ whose columns span the corresponding p-dimensional subspace in $\mathbb{R}^{|\mathcal{N}(v_i)| \times p}$. The distance between such subspaces can be computed as a set of principal angles $\{\theta_i\}_{i=1}^{k}$ between these subspaces. (Dong et al., 2013) showed that the projection distance between two subspaces $\mathbf{Y}_1$ and $\mathbf{Y}_2$ can be represented as a separate trace minimization problem:

$$d_{\text{proj}}^2(\mathbf{Y}_1, \mathbf{Y}_2) = \sum_{i=1}^{p} \sin^2 \theta_i = k - \operatorname{tr}(\mathbf{Y}_1 \mathbf{Y}_1^\top \mathbf{Y}_2 \mathbf{Y}_2^\top). \tag{22}$$

Based on this, we further define the projection of the final representative subspace $\mathbf{U}$ and the $k$ sparse candidate subspace $\{\mathbf{U}_m\}_{m=1}^{k}$:

$$d_{\text{proj}}^2(\mathbf{U}, \{\mathbf{U}_m\}_{m=1}^{k}) = \sum_{m=1}^{k} d_{\text{proj}}^2(\mathbf{U}, \mathbf{U}_m) = p \times k - \sum_{m=1}^{k} \operatorname{tr}(\mathbf{U}\mathbf{U}^\top \mathbf{U}_m \mathbf{U}_m^\top), \tag{23}$$

which ensures that individual subspaces are close to the final representative subspace $\mathbf{U}$.

Finally, to maintain the original vertex connectivity from all $k$ sparse ego-graphs and emphasize the connectivity relationship from more reliable sparsifiers (with higher expert scores), we propose the

following objective function:

$$\min_{\mathbf{U}_m \in \mathbb{R}^{|\mathcal{N}(v_m)| \times p}} \sum_{m=1}^{k} E_m^{(i)} \left( p \times k - \sum_{m=1}^{k} \text{tr} \left( \mathbf{U}\mathbf{U}^\top \mathbf{U}_m \mathbf{U}_m^\top \right) \right), \tag{24}$$

where $E_m^{(i)}$ represents the expert score of the node $v_i$'s $m$-th sparsifier expert. Based on Equations (21) and (24), we present the overall objective:

$$\min_{\mathbf{U}^{(i)} \in \mathbb{R}^{|\mathcal{N}(v_i)| \times p}} \sum_{m=1}^{k} \left( \underbrace{\text{tr}(\mathbf{U}^{(i)^\top} \mathbf{L}_m \mathbf{U}^{(i)})}_{\text{(1) node connectivity}} + \underbrace{E_m^{(i)} \cdot d^2(\mathbf{U}^{(i)}, \mathbf{U}_m^{(i)})}_{\text{(2) subspace distance}} \right), \text{s. t. } \mathbf{U}^{(i)^\top} \mathbf{U}^{(i)} = \mathbf{I}. \tag{25}$$

For simplicity, we omit the superscript $^{(i)}$ in the following content. Substituting Equation (23) into Equation (25), we obtain:

$$\min_{\mathbf{U}} \sum_{m=1}^{k} \text{tr}(\mathbf{U}^T \mathbf{L}_m \mathbf{U}) + E_m \cdot \left( p \times k - \sum_{m=1}^{k} \text{tr}(\mathbf{U}\mathbf{U}^T \mathbf{U}_m \mathbf{U}_m^T) \right), \text{s.t.} \mathbf{U}^T \mathbf{U} = \mathbf{I}. \tag{26}$$

Further simplification by neglecting constant terms like $E_m \times p \times k$ yields:

$$\min_{\mathbf{U}} \sum_{m=1}^{k} \text{tr}(\mathbf{U}^T \mathbf{L}_m \mathbf{U}) - E_m \cdot \sum_{m=1}^{k} \text{tr}(\mathbf{U}\mathbf{U}^T \mathbf{U}_m \mathbf{U}_m^T), \text{s.t.} \mathbf{U}^T \mathbf{U} = \mathbf{I}. \tag{27}$$

Reorganizing the trace form of the second term, we obtain:

$$\min_{\mathbf{U}} \text{tr} \left[ \mathbf{U}^T (\sum_{k=1}^{M} \mathbf{L}_m - E_m \sum_{k=1}^{M} \mathbf{U}_m \mathbf{U}_m^T) \mathbf{U} \right], \text{s.t.} \mathbf{U}^T \mathbf{U} = \mathbf{I}. \tag{28}$$

At this point, the optimization problem essentially becomes a trace minimization problem, and thus the solution to this minimization problem is essentially the term between $\mathbf{U}^T$ and $\mathbf{U}$, which is:

$$\widehat{\mathbf{L}} = (\sum_{k=1}^{M} \mathbf{L}_m - E_m \sum_{k=1}^{M} \mathbf{U}_m \mathbf{U}_m^T) = \sum_{k=1}^{M} (\mathbf{L}_m - E_m \cdot \mathbf{U}_m \mathbf{U}_m^T). \tag{29}$$

Since computations involving the Grassmann manifold unavoidably entail eigenvalue decomposition, concerns about computational complexity may arise. However, given that MoG only operates mixtures on the ego-graph of each node, such computational burden is entirely acceptable. Specific complexity analyses are presented in Appendix E.

## C.1 FURTHER DISCUSSION ON THE USE OF GRASSMANN MANIFOLD

In this section, we visualize the impact of different ego-graph ensemble methods on the eigenvalue distribution. Specifically, we select node [2458] from the OGBN-ARXIV dataset and compare its original ego-graph, the sparse ego-graphs produced by three different sparsifiers, and the ensembled ego-graphs obtained through simple averaging and Grassmann optimization. The eigenvalue distributions are shown in Figure 5. As observed, when the three candidate sparse graphs are combined through simple averaging followed by sparsification, the resulting graph's eigenvalue distribution can deviate significantly from the original distribution. In contrast, the Grassmann ensembling method effectively preserves the spectral properties of each graph view, resulting in a sparse ego-graph whose eigenvalue distribution closely aligns with that of the original graph.

## D ALGORITHM WORKFLOW

The algorithm framework is presented in Algo. 1.

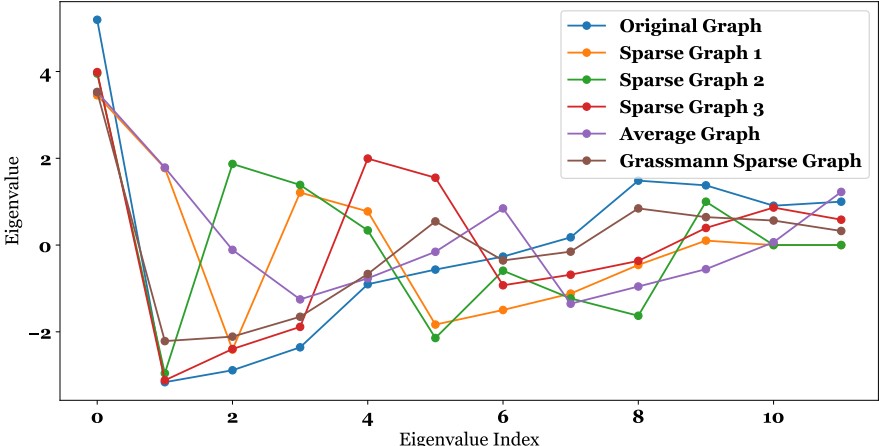

Figure 5: Case study on the eigenvalue distribution of ego-graph.

## E  COMPLEXITY ANALYSIS

In this section, we delve into a comprehensive analysis of the time and space complexity of MoG. Without loss of generality, we consider the scenario where MoG is applied to vanilla GCN. It is worth recalling that the forward time complexity of vanilla GCN is given by:

$$\mathcal{O}(L \times |\mathcal{E}| \times D + L \times |\mathcal{V}| \times D^2), \tag{30}$$

where $L$ is the number of GNN layers, $|\mathcal{E}|$ and $|\mathcal{V}|$ denotes the number of edges and nodes, respectively, and $D$ is the hidden dimension. Similarly, the forward space complexity of GCN is:

$$\mathcal{O}(L \times |\mathcal{E}| + L \times D^2 + L \times |\mathcal{V}| \times D) \tag{31}$$

When MoG is applied to GCN, each sparsifier expert $\kappa(\cdot)$ essentially introduces additional complexity equivalent to that of an $\mathrm{FFN}(\cdot)$, as depicted in Equation (9). Incorporating the Sparse MoE-style structure, the forward time complexity of GCN+MoG becomes:

$$\mathcal{O}(L \times |\mathcal{E}| \times D + L \times |\mathcal{V}| \times D^2 + k \times |\mathcal{E}| \times D \times D^s), \tag{32}$$

where $D^s$ represents the hidden dimension of the feed-forward network in Equation (9) and $k$ denotes the number of selected experts. Similarly, the forward space complexity is increased to:

$$\mathcal{O}(L \times |\mathcal{E}| + L \times D^2 + L \times |\mathcal{V}| \times D + k \times |\mathcal{E}| \times D \times D^s). \tag{33}$$

It is noteworthy that we omit the analysis for the routing network, as its computational cost is meanwhile negligible compared to the cost of selected experts, since both $W_g \in \mathbb{R}^{K \times F}$ and $W_n \in \mathbb{R}^{K \times F}$ is in a much smaller dimension that the weight matrix $W \in \mathbb{R}^{F \times F}$ in GCN.

Furthermore, we present the additional complexity introduced by the step of graph mixture on the Grassmann manifold. For each center node's $k$ sparse ego-graphs, we need to compute the graph Laplacian and the eigenvector matrix, which incurs an extra time complexity of $\mathcal{O}(k \times (\frac{|\mathcal{E}|}{|\mathcal{V}|})^3)$; to compute the Laplacian $\widehat{\mathbf{L}^{(i)}}$ of the final ensemble sparse graph, an additional complexity of $\mathcal{O}(k \times (\frac{|\mathcal{E}|}{|\mathcal{V}|})^2 \times p)$ is required. In the end, the complete time complexity of MoG is expressed as:

$$\mathcal{O}\left(\underbrace{L \times |\mathcal{E}| \times D + L \times |\mathcal{V}| \times D^2}_{\text{vanilla GCN}} + \underbrace{k \times |\mathcal{E}| \times D \times D^s}_{\text{sparsifier experts}} + \underbrace{k \left(\frac{|\mathcal{E}|}{|\mathcal{V}|}\right)^3 + k \left(\frac{|\mathcal{E}|}{|\mathcal{V}|}\right)^2 p}_{\text{graph mixture}}\right). \tag{34}$$

To empirically verify that MoG does not impose excessive computational burdens on GNN backbones, we conduct experiments in Section 4.5 to compare the per-epoch time efficiency metric of MoG with other sparsifiers.

---

**Algorithm 1:** Algorithm workflow of MoG

---

**Input** : $\mathcal{G} = (\mathbf{A}, \mathbf{X})$, GNN model $f(\mathcal{G}, \boldsymbol{\Theta})$, , epoch number $Q$.
**Output** : Sparse graph $\mathcal{G}^{\text{sub}} = \{\mathcal{V}, \mathcal{E}'\}$
**for** iteration $q \leftarrow 1$ **to** $Q$ **do**
    /* `Ego-graph decomposition`                                         */
    Decompose $\mathcal{G}$ into ego-graph representations $\{\mathcal{G}^{(1)}, \mathcal{G}^{(2)}, \cdots, \mathcal{G}^{(N)}\}$.
    /* `Sparsifier expert allocation`                                   */
    **for** node $i \leftarrow 1$ **to** $|\mathcal{V}|$ **do**
       | Calculate the total $K$ expert score of $v_i$ by routing network $\psi(x_i)$;         ▷ Eq. 3
       | Select $k$ sparsifier expert for node $v_i$ by $\text{Softmax}(\text{TopK}(\psi(x_i), k))$;         ▷ Eq. 3
    **end**
    /* `Produce sparse graph condidates`                              */
    **for** iteration $i \leftarrow 1$ **to** $|\mathcal{V}|$ **do**
       **for** sparsifier index $m \leftarrow 1$ **to** $m$ **do**
          | Sparsifier $\kappa^m$ determines which edges to remove by
          | $\mathcal{E}_p^{(i)} = \text{TopK}\left(-C^m(\mathcal{E}), \lceil |\mathcal{E}^{(i)}| \times s\% \rceil\right)$;         ▷ Eq. 8
          | Produce sparse graph candidate $\widetilde{\mathcal{G}}^{(i)} = \kappa^m(\mathcal{G}^{(i)}) = \{\mathcal{V}^{(i)}, \mathcal{E}^{(i)} \setminus \mathcal{E}_p^{(i)}\}$.
       **end**
       /* `Ensenmble sparse graphs on Grassmann manifold`               */
       Calculate the ensemble graph's graph Laplacian by
       $\widehat{\mathbf{L}^{(i)}} = \sum_{m=1}^{k} \left(\mathbf{L}_m - E_m^{(i)} \cdot {\mathbf{U}^{(\mathbf{i})}}^\top \mathbf{U}^{(\mathbf{i})}\right)$;         ▷ Eq. 12
       Obtain $v_i$'s final sparse graph by $\text{ESM}(\{\widehat{\mathcal{G}^{(i)}} = \{\mathbf{D} - \widehat{\mathbf{L}^{(i)}}, \mathbf{X}^{(i)}\})$;         ▷ Eq. 13
       Compute $v_i$'s weighted sparsity by $s^{(i)}\% = \frac{1}{k} \sum_{m=1}^{k} s^m\%$;         ▷ Eq. 14
       Post-sparsify $\widehat{\mathcal{G}^{(i)}}$: $\widehat{\mathcal{G}^{(i)}} \leftarrow \{\text{TopK}(\widehat{\mathbf{A}^{(i)}}, |\mathcal{E}^{(i)}| \times s^{(i)}\%), \mathbf{X}^{(i)}\}$;         ▷ Eq. 14
    **end**
    /* `Combine ego-graphs`                                              */
    $\widehat{\mathcal{G}} \leftarrow \{\widehat{\mathcal{G}^{(1)}}, \widehat{\mathcal{G}^{(2)}}, \cdots, \widehat{\mathcal{G}^{(|\mathcal{V}|)}}\}$
    /* `Standard GNN training`                                         */
    Feed the sparse graph $\widehat{\mathcal{G}}$ into GNN model for any kinds of downstream training ; ▷ Eq. 6
    Compute loss $\mathcal{L}_{\text{task}} + \lambda \cdot \mathcal{L}_{\text{importance}}$;         ▷ Eq. 16
    Backpropagate to update GNN $f(\mathcal{G}, \boldsymbol{\Theta})$, routing network $\psi$ and sparsifiers $\{\kappa^m\}_{m=1}^{K}$.
**end**

---

# F EXPERIMENTAL DETAILS

## F.1 DATASET STATISTICS

We conclude the dataset statistics in Tab. 7

Table 7: Graph datasets statistics.

| Dataset | #Graph | #Node | #Edge | #Classes | Metric |
|---|---|---|---|---|---|
| OGBN-ARXIV | 1 | 169,343 | 1,166,243 | 40 | Accuracy |
| OGBN-PROTEINS | 1 | 132,534 | 39,561,252 | 2 | ROC-AUC |
| OGBN-PRODUCTS | 1 | 2,449,029 | 61,859,140 | 47 | Accuracy |
| OGBG-PPA | 158,100 | 243.4 | 2,266.1 | 47 | Accuracy |
| MNIST | 70,100 | 50.5 | 564.5 | 10 | Accuracy |
| CIFAR-10 | 60,000 | 117.6 | 914.0 | 10 | Accuracy |

## F.2 Evaluation Metrics

Accuracy represents the ratio of correctly predicted outcomes to the total predictions made. The ROC-AUC (Receiver Operating Characteristic-Area Under the Curve) value quantifies the probability that a randomly selected positive example will have a higher rank than a randomly selected negative example.

## F.3 Dataset Splits

For **node-level tasks**, the data splits for OGBN-ARXIV, OGBN-PROTEINS, and OGBN-PRODUCTS were provided by the benchmark (Hu et al., 2020). Specifically, for OGBN-ARXIV, we train on papers published until 2017, validate on papers from 2018 and test on those published since 2019. For OGBN-PROTEINS, protein nodes were segregated into training, validation, and test sets based on their species of origin. For OGBN-PRODUCTS, we sort the products according to their sales ranking and use the top 8% for training, next top 2% for validation, and the rest for testing.

For **graph-level tasks**, we follow (Hu et al., 2020) for OGBG-PPA. Concretely, we adopt the species split, where the neighborhood graphs in the validation and test sets are extracted from protein association networks of species not encountered during training but belonging to one of the 37 taxonomic groups. This split stress-tests the model's capacity to extract graph features crucial for predicting taxonomic groups, enhancing biological understanding of protein associations. For MNIST and CIFAR-10, consistent with (Dwivedi et al., 2020), we split them to 55000 train/5000 validation/10000 test for MNIST, and 45000 train/5000 validation/10000 test for CIFAR10, respectively. We report the test accuracy at the epoch with the best validation accuracy.

## F.4 Parameter Setting

**Backbone Parameters** For node classification backbones, we utilize a 3-layer Graph-SAGE with `hidden_dim` $\in \{128, 256\}$. As for DeeperGCN, we set `layer_num` = 28, `block` = `res+`, `hidden_dim` = 64. The other configurations are the same as in `https://github.com/lightaime/deep_gcns_torch/tree/master/examples/ogb/ogbn_proteins`. For graph classification backbones, we leverage a 4-layer PNA with `hidden_dim` = 300. Rest configurations are the same as in `https://github.com/lukecavabarrett/pna`.

**MoG parameters** We adopt the $m = 4$ sparsity criteria outlined in Section 3.3, assigning $n = 3$ different sparsity levels $\{s_1, s_2, s_3\}$ to each criterion, resulting in a total of $K = m \times n = 12$ experts. We select $k = 2$ sparsifier experts for each node, and set the loss scaling factor $\lambda = 1e - 2$ across all datasets and backbones.

All the experiments are conducted on NVIDIA Tesla V100 (32GB GPU), using PyTorch and PyTorch Geometric framework.

## F.5 Sparsifier Baseline Configurations

- Topology-based sparsification

  - **Rank Degree** (Talati et al., 2022): The Rank Degree sparsifier initiates by selecting a random set of "seed" vertices. Then, the vertices with connections to these seed vertices are ranked based on their degree in descending order. Subsequently, the edges linking each seed vertex to its top-ranked neighbors are chosen and integrated into the sparsified graph. The newly added nodes in the graph act as new seeds for identifying additional edges. This iterative process continues until the target sparsification limit is attained. We utilize the implementation in (Chen et al., 2023).

  - **Local Degree** (Hamann et al., 2016): Local Degree sparsifier, similar to Rank Degree, incorporates edges to the top $\deg(v)^\alpha$ neighbors ranked by their degree in descending order, where $\alpha \in [0, 1]$ represents the degree of sparsification.

  - **Forest Fire** (Leskovec et al., 2006): Forest fire assembles "burning" through edges probabilistically, and we use the implementation in (Staudt et al., 2016).

- **G-Spar** (Murphy, 1996b): G-Spar sorts the Jaccard scores globally and then selects the edges with the highest similarity score. We opt for the code from (Staudt et al., 2016).
- **Local Similarity** (Satuluri et al., 2011a): Local Similarity ranks edges using the Jaccard score and computes $\log(\mathrm{rank}(e_{ij}))/\log(\deg(e_{ij}))$ as the similarity score, and selects edges with the highest similarity scores. We utilize the implementation in (Chen et al., 2023).
- **SCAN** (Spielman & Srivastava, 2008): SCAN uses structural similarity (called SCAN similarity) measures to detect clusters, hubs, and outliers. We utilize the implementation in (Chen et al., 2023)
- **DSpar** (Liu et al., 2023b): DSpar is an extension of effective resistance sparsifier, which aims to reduce the high computational budget of calculating effective resistance through an unbiased approximation. We adopt their official implementation (Liu et al., 2023b).

- Semantic-based sparsification

  - **UGS** (Chen et al., 2021): We utilize the official implementation from the authors. Notably, UGS was originally designed for joint pruning of model parameters and edges. Specifically, it sets separate pruning parameters for parameters and edges, namely the weight pruning ratio $p_\theta$ and the graph pruning ratio $p_g$. In each iteration, a corresponding proportion of parameters/edges is pruned. For a fairer comparison, we set $p_\theta = 0\%$, while maintaining $p_g \in \{5\%, 10\}$ (consistent with the original paper).
  - **GEBT** (You et al., 2022): GEBT, for the first time, discovered the existence of graph early-bird (GEB) tickets that emerge at the very early stage when sparsifying GCN graphs. (You et al., 2022) has proposed two variants of graph early bird tickets, and we opt for the graph-sparsification-only version, dubbed GEB Ticket Identification.
  - **Meta-gradient sparsifier** (Wan & Schweitzer, 2021): The Meta-gradient sparsifier prunes edges based on their meta-gradient importance scores, assessed over multiple training epochs. Since no official implementation is provided, we carefully replicated the results following the guidelines in the original paper.
  - **ACE-GLT** (Wang et al., 2023b): ACE-GLT inherits the iterative magnitude pruning (IMP) paradigm from UGS. Going beyond UGS, it suggested mining valuable information from pruned edges/weights after each round of IMP, which in the meanwhile doubled the computational cost of IMP. We utilize the official implementation provided by (Wang et al., 2023b), and set $p_\theta = 0\%, p_g \in \{5\%, 10\}$.
  - **WD-GLT** (Hui et al., 2023): WD-GLT also inherits the iterative magnitude pruning paradigm from UGS, so we also set $p_\theta = 0\%, p_g \in \{5\%, 10\%\}$ across all datasets and backbones. The perturbation ratio $\alpha$ is tuned among $\{0, 1\}$. Since no official implementation is provided, we carefully reproduced the results according to the original paper.
  - **AdaGLT** (Zhang et al., 2024a): AdaGLT revolutionizes the original IMP-based graph lottery ticket methodology into an adaptive, dynamic, and automated approach, proficient in identifying sparse graphs with layer-adaptive structures. We fix $\eta_\theta = 0\%, \eta_g \in \{1e-6, 1e-5, 1e-4, 1e-3, 1e-2\}, \omega = 2$ across all datasets and backbones.

### F.6 ADJUSTING GRAPH SPARSITY

In Table 8, we provide detailed guidelines on how to achieve the desired global sparsity by adjusting the three sparsity levels $\{s_1, s_2, s_3\}$ in MoG across six datasets.

## G ADDTIONAL EXPERIMENT RESULTS

### G.1 RESULTS FOR RQ1

We report the performances of MoG and other sparsifiers on OGBN-PRODUCTS in Table 9.

### G.2 SENSITIVITY ANALYSIS OF PARAMETER K

Based on the experiments in Section 4.4, we further provide sensitivity analysis results on OGBN-PROTEINS+RevGNN, as shown in Table 13. It can be observed that MoG achieves peak performance at $k \in \{2, 3\}$ and begins to decline after $k \geq 4$, which is consistent with our finding in Observation 6.

Table 8: The recipe for adjusting graph sparsity via different sparsifier combinations.

| Datasets | $1 - s_1$ | $1 - s_2$ | $1 - s_3$ | $k$ | $1 - s\%$ |
|---|---|---|---|---|---|
| OGBN-ARXIV | 1 | 0.9 | 0.8 | 2 | [88.0%,90.9%] |
| | 0.8 | 0.7 | 0.5 | 2 | [69.0%,73.2%] |
| | 0.6 | 0.5 | 0.3 | 2 | [49.5%,52.7%] |
| | 0.5 | 0.3 | 0.15 | 2 | [27.1%, 31.6%] |
| OGBN-PROTEINS | 1 | 0.9 | 0.8 | 2 | [86.1%,89.3%] |
| | 0.8 | 0.7 | 0.6 | 2 | [65.1%,69.2%] |
| | 0.6 | 0.5 | 0.4 | 2 | [45.2%,49.3%] |
| | 0.4 | 0.3 | 0.2 | 2 | [29.2%,31.1%] |
| OGBN-PRODUCTS | 1 | 0.9 | 0.8 | 2 | [90.1%,93.2%] |
| | 0.8 | 0.7 | 0.6 | 2 | [69.3%,72.0%] |
| | 0.6 | 0.5 | 0.4 | 2 | [51.5%,54.9%] |
| | 0.4 | 0.3 | 0.2 | 2 | [28.7%,36.0%] |
| MNIST | 1 | 0.85 | 0.8 | 2 | [90.4%,92.7%] |
| | 0.8 | 0.5 | 0.4 | 2 | [67.1%,68.3%] |
| | 0.6 | 0.3 | 0.2 | 2 | [46.2%,49.3%] |
| | 0.35 | 0.1 | 0.1 | 2 | [29.8%,31.3%] |
| CIFAR-10 | 1 | 0.85 | 0.8 | 2 | [90.6%,93.7%] |
| | 0.8 | 0.5 | 0.4 | 2 | [67.5%,69.9%] |
| | 0.6 | 0.3 | 0.2 | 2 | [47.7%,49.3%] |
| | 0.35 | 0.1 | 0.1 | 2 | [30.1%,31.3%] |
| OGBG-PPA | 0.95 | 0.9 | 0.8 | 2 | [86.5%,88.9%] |
| | 0.8 | 0.65 | 0.6 | 2 | [68.0%,70.1%] |
| | 0.6 | 0.5 | 0.3 | 2 | [47.8%,48.9%] |
| | 0.4 | 0.3 | 0.15 | 2 | [30.1%,33.6%] |

Table 9: Node classification performance comparison to state-of-the-art sparsification methods. All methods are trained using **GraphSAGE**, and the reported metrics represent the average of **five runs**. We denote methods with † that do not have precise control over sparsity; their performance is reported around the target sparsity $\pm 2\%$. We do not report results for sparsifiers like ER for OOT issues and those like UGS for their infeasibility in inductive settings (mini-batch training).

| | Dataset | OGBN-PRODUCTS (Accuracy ↑) | | | |
|---|---|---|---|---|---|
| | Sparsity % | 10 | 30 | 50 | 70 |
| Topology | Random | $76.99_{\downarrow 1.05}$ | $74.21_{\downarrow 3.83}$ | $71.08_{\downarrow 6.96}$ | $67.24_{\downarrow 10.80}$ |
| | Rank Degree† (Voudigari et al., 2016) | $76.08_{\downarrow 1.96}$ | $74.26_{\downarrow 3.89}$ | $71.85_{\downarrow 6.19}$ | $70.66_{\downarrow 7.38}$ |
| | Local Degree† (Hamann et al., 2016) | $77.19_{\downarrow 1.58}$ | $76.40_{\downarrow 1.64}$ | $72.77_{\downarrow 5.27}$ | $72.48_{\downarrow 5.56}$ |
| | G-Spar (Murphy, 1996b) | $76.15_{\downarrow 1.89}$ | $74.20_{\downarrow 3.84}$ | $71.55_{\downarrow 6.49}$ | $69.42_{\downarrow 8.62}$ |
| | LSim† (Satuluri et al., 2011a) | $77.96_{\downarrow 0.08}$ | $74.98_{\downarrow 2.06}$ | $72.67_{\downarrow 5.37}$ | $70.43_{\downarrow 7.61}$ |
| | SCAN (Xu et al., 2007) | $76.30_{\downarrow 1.74}$ | $74.33_{\downarrow 3.71}$ | $71.25_{\downarrow 6.79}$ | $71.12_{\downarrow 6.92}$ |
| | DSpar (Liu et al., 2023b) | $78.25_{\uparrow 0.21}$ | $75.11_{\downarrow 2.93}$ | $74.57_{\downarrow 3.47}$ | $73.16_{\downarrow 4.88}$ |
| Sema | AdaGLT (Zhang et al., 2024a) | $78.19_{\uparrow 0.15}$ | $77.30_{\downarrow 0.74}$ | $74.38_{\downarrow 3.66}$ | $73.04_{\downarrow 5.00}$ |
| | **MoG (Ours)**† | $\mathbf{78.77}_{\uparrow 0.73}$ | $\mathbf{78.15}_{\uparrow 0.11}$ | $\mathbf{76.98}_{\downarrow 1.06}$ | $\mathbf{74.91}_{\downarrow 3.17}$ |
| | Whole Dataset | $78.04_{\pm 0.31}$ | | | |

Table 10: Node classification performance comparison to state-of-the-art sparsification methods. All methods are trained using **DeeperGCN**, and the reported metrics represent the average of **five runs**. We denote methods with † that do not have precise control over sparsity; their performance is reported around the target sparsity $\pm 2\%$. "OOM" and "OOT" denotes out-of-memory and out-of-time, respectively.

| | Dataset | OGBN-ARXIV (Accuracy ↑) | | | |
|---|---|---|---|---|---|
| | Sparsity % | 10 | 30 | 50 | 70 |
| Topology-guided | Random | $70.66_{\downarrow 1.28}$ | $68.74_{\downarrow 3.20}$ | $65.38_{\downarrow 6.56}$ | $63.55_{\downarrow 8.39}$ |
| | Rank Degree† (Voudigari et al., 2016) | $69.44_{\downarrow 2.50}$ | $67.82_{\downarrow 4.12}$ | $65.08_{\downarrow 6.86}$ | $63.19_{\downarrow 8.75}$ |
| | Local Degree† (Hamann et al., 2016) | $68.77_{\downarrow 3.17}$ | $67.92_{\downarrow 4.02}$ | $66.10_{\downarrow 5.84}$ | $65.97_{\downarrow 5.97}$ |
| | Forest Fire† (Leskovec et al., 2006) | $68.70_{\downarrow 3.24}$ | $68.95_{\downarrow 3.99}$ | $67.23_{\downarrow 4.71}$ | $67.29_{\downarrow 4.65}$ |
| | G-Spar (Murphy, 1996b) | $70.57_{\downarrow 1.37}$ | $70.15_{\downarrow 1.79}$ | $68.77_{\downarrow 3.17}$ | $65.26_{\downarrow 6.68}$ |
| | LSim† (Satuluri et al., 2011a) | $69.33_{\downarrow 2.61}$ | $67.19_{\downarrow 4.75}$ | $63.55_{\downarrow 8.39}$ | $62.20_{\downarrow 9.74}$ |
| | SCAN (Xu et al., 2007) | $71.33_{\downarrow 0.61}$ | $69.22_{\downarrow 2.72}$ | $67.88_{\downarrow 4.06}$ | $64.32_{\downarrow 7.62}$ |
| | ER (Spielman & Srivastava, 2008) | $71.33_{\downarrow 0.61}$ | $69.65_{\downarrow 2.29}$ | $69.08_{\downarrow 2.86}$ | $67.10_{\downarrow 4.84}$ |
| | DSpar (Liu et al., 2023b) | $71.65_{\downarrow 0.29}$ | $70.66_{\downarrow 1.28}$ | $68.03_{\downarrow 3.91}$ | $67.25_{\downarrow 4.69}$ |
| Semantic-guided | UGS† (Chen et al., 2021) | $72.01_{\uparrow 0.93}$ | $70.29_{\downarrow 1.65}$ | $68.43_{\downarrow 3.51}$ | $67.85_{\downarrow 4.09}$ |
| | GEBT (You et al., 2022) | $70.22_{\downarrow 1.72}$ | $69.40_{\downarrow 2.54}$ | $67.84_{\downarrow 4.10}$ | $67.49_{\downarrow 4.45}$ |
| | MGSpar (Wan & Schweitzer, 2021) | $70.02_{\downarrow 1.92}$ | $69.34_{\downarrow 2.60}$ | $68.02_{\downarrow 3.92}$ | $65.78_{\downarrow 6.16}$ |
| | ACE-GLT† (Wang et al., 2023b) | $72.13_{\uparrow 0.19}$ | $71.96_{\uparrow 0.02}$ | $69.13_{\downarrow 2.81}$ | $67.93_{\downarrow 4.01}$ |
| | WD-GLT† (Hui et al., 2023) | $71.92_{\downarrow 0.02}$ | $70.21_{\downarrow 1.73}$ | $68.30_{\downarrow 3.64}$ | $66.57_{\downarrow 5.37}$ |
| | AdaGLT (Zhang et al., 2024a) | $71.98_{\uparrow 0.04}$ | $70.44_{\downarrow 1.50}$ | $69.15_{\downarrow 2.79}$ | $68.05_{\downarrow 3.89}$ |
| | **MoG (Ours)**† | $\mathbf{72.08}_{\uparrow 0.14}$ | $\mathbf{71.98}_{\uparrow 0.05}$ | $\mathbf{69.86}_{\downarrow -2.08}$ | $\mathbf{68.20}_{\downarrow -3.74}$ |
| | Whole Dataset | $71.93_{\pm 0.04}$ | | | |

Table 11: Node classification performance comparison to state-of-the-art sparsification methods. All methods are trained using **DeeperGCN**, and the reported metrics represent the average of **five runs**. We denote methods with † that do not have precise control over sparsity; their performance is reported around the target sparsity $\pm 2\%$. "OOM" and "OOT" denotes out-of-memory and out-of-time, respectively.

| | Dataset | OGBN-PROTEINS (ROC-AUC ↑) | | | |
|---|---|---|---|---|---|
| | Sparsity % | 10 | 30 | 50 | 70 |
| Topology-guided | Random | $80.18_{\downarrow 2.55}$ | $78.92_{\downarrow 3.83}$ | $76.57_{\downarrow 6.16}$ | $72.69_{\downarrow 10.04}$ |
| | Rank Degree† (Voudigari et al., 2016) | $80.14_{\downarrow 2.59}$ | $79.05_{\downarrow 3.73}$ | $78.59_{\downarrow 4.13}$ | $76.22_{\downarrow 6.51}$ |
| | Local Degree† (Hamann et al., 2016) | $79.40_{\downarrow 3.33}$ | $79.83_{\downarrow 3.90}$ | $78.50_{\downarrow 4.23}$ | $78.25_{\downarrow 4.48}$ |
| | Forest Fire† (Leskovec et al., 2006) | $81.49_{\downarrow 1.24}$ | $78.47_{\downarrow 4.26}$ | $76.14_{\downarrow 6.59}$ | $73.89_{\downarrow 9.84}$ |
| | G-Spar (Murphy, 1996b) | $81.56_{\downarrow 1.17}$ | $81.12_{\downarrow 1.61}$ | $79.13_{\downarrow 3.60}$ | $77.45_{\downarrow 5.28}$ |
| | LSim† (Satuluri et al., 2011a) | $80.30_{\downarrow 2.43}$ | $79.19_{\downarrow 3.54}$ | $77.13_{\downarrow 5.60}$ | $77.85_{\downarrow 4.88}$ |
| | SCAN (Xu et al., 2007) | $81.60_{\downarrow 1.13}$ | $80.19_{\downarrow 2.54}$ | $81.53_{\downarrow 1.20}$ | $78.58_{\downarrow 4.15}$ |
| | ER (Spielman & Srivastava, 2008) | OOT | | | |
| | DSpar (Liu et al., 2023b) | $81.46_{\downarrow 1.27}$ | $80.57_{\downarrow 2.16}$ | $77.41_{\downarrow 5.32}$ | $75.35_{\downarrow 7.39}$ |
| Semantic-guided | UGS† (Chen et al., 2021) | $82.33_{\downarrow 0.40}$ | $81.54_{\downarrow 1.19}$ | $78.75_{\downarrow 4.98}$ | $76.40_{\downarrow 6.33}$ |
| | GEBT (You et al., 2022) | $80.74_{\downarrow 2.99}$ | $80.22_{\downarrow 2.51}$ | $79.81_{\downarrow 3.92}$ | $76.05_{\downarrow 6.68}$ |
| | MGSpar (Wan & Schweitzer, 2021) | OOM | | | |
| | ACE-GLT† (Wang et al., 2023b) | $82.93_{\uparrow 0.80}$ | $82.01_{\downarrow 0.72}$ | $81.05_{\downarrow 1.68}$ | $75.92_{\downarrow 6.81}$ |
| | WD-GLT† (Hui et al., 2023) | OOM | | | |
| | AdaGLT (Zhang et al., 2024a) | $82.60_{\downarrow 0.13}$ | $82.76_{\uparrow 0.97}$ | $80.55_{\downarrow 2.18}$ | $78.42_{\downarrow 4.31}$ |
| | **MoG (Ours)**† | $\mathbf{83.32}_{\uparrow 0.41}$ | $\mathbf{82.14}_{\downarrow 0.59}$ | $\mathbf{81.92}_{\downarrow 0.81}$ | $\mathbf{80.90}_{\downarrow 1.83}$ |
| | Whole Dataset | $82.73_{\pm 0.02}$ | | | |

Table 12: Graph classification performance comparison to state-of-the-art sparsification methods. All methods are trained using **PNA**, and the reported metrics represent the average of **five runs**. We denote methods with † that do not have precise control over sparsity; their performance is reported around the target sparsity $\pm 2\%$.

| | Dataset | CIFAR-10 (Accuracy ↑) | | | |
|---|---|---|---|---|---|
| | Sparsity % | 10 | 30 | 50 | 70 |
| Topology | Random | $68.04_{\downarrow 1.70}$ | $66.81_{\downarrow 2.93}$ | $65.35_{\downarrow 4.39}$ | $62.14_{\downarrow 7.60}$ |
| | Rank Degree† (Voudigari et al., 2016) | $68.27_{\downarrow 1.77}$ | $67.14_{\downarrow 2.60}$ | $64.05_{\downarrow 5.69}$ | $60.22_{\downarrow 9.52}$ |
| | Local Degree† (Hamann et al., 2016) | $68.10_{\downarrow 1.64}$ | $67.29_{\downarrow 2.45}$ | $64.96_{\downarrow 4.78}$ | $61.77_{\downarrow 8.97}$ |
| | G-Spar (Murphy, 1996b) | $67.13_{\downarrow 2.61}$ | $65.06_{\downarrow 4.68}$ | $64.86_{\downarrow 4.88}$ | $62.92_{\downarrow 6.82}$ |
| | LSim† (Satuluri et al., 2011a) | $69.75_{\uparrow 0.01}$ | $67.33_{\downarrow 2.41}$ | $66.58_{\downarrow 3.16}$ | $64.86_{\downarrow 4.88}$ |
| | SCAN (Xu et al., 2007) | $68.25_{\downarrow 1.49}$ | $66.11_{\downarrow 3.63}$ | $64.59_{\downarrow 5.15}$ | $63.20_{\downarrow 6.54}$ |
| | DSpar (Liu et al., 2023b) | $68.94_{\uparrow 0.53}$ | $66.80_{\downarrow 2.94}$ | $64.87_{\downarrow 4.87}$ | $64.10_{\downarrow 5.64}$ |
| Sema | AdaGLT (Zhang et al., 2024a) | $69.77_{\uparrow 0.02}$ | $67.97_{\downarrow 1.78}$ | $65.06_{\downarrow 4.68}$ | $64.22_{\downarrow 5.52}$ |
| | **MoG (Ours)**† | $\mathbf{70.04}_{\uparrow 0.30}$ | $\mathbf{69.80}_{\uparrow -0.94}$ | $\mathbf{68.28}_{\downarrow -1.46}$ | $\mathbf{66.55}_{\downarrow -3.19}$ |
| | Whole Dataset | | $69.74_{\pm 0.17}$ | | |

Table 13: Sensitivity analysis of parameter $k$ when applying MoG to OGBN-PROTEINS+RevGNN.

| $k$ | 1 | 2 | 3 | 4 | 5 |
|---|---|---|---|---|---|
| MoG | 88.37 | 89.04 | 89.09 | 88.55 | 88.20 |

We also reported the impact of selecting different values of $k$ on the per-epoch training time and inference time, when applying MoG to OGBN-ARXIV+GraphSAGE in Table 14. It can be observed that although the training and inference cost of MoG increases as the number of selected experts increases, this additional cost is not significant: when $k$ doubles from 2 to 4, the inference time only increases by 23%. More importantly, we can already achieve optimal performance with $k \in \{2, 3\}$, so there is no need to select too many experts, therefore avoiding significant inference delay.

Table 14: Sensitivity analysis of parameter $k$ when applying MoG to OGBN-ARXIV+GraphSAGE.

| Sparsity | Selected Expert $k$ | Per-Epoch Time | Inference Time | Acc. |
|---|---|---|---|---|
| 30% | 2 | 0.213 | 0.140 | 70.53 |
| 30% | 3 | 0.241 | 0.161 | 70.48 |
| 30% | 4 | 0.266 | 0.173 | 70.13 |
| 30% | 5 | 0.279 | 0.190 | 69.57 |

### G.3 ABLATION STUDY ON THE NOISE CONTROL OF THE ROUTER NETWORK

We test three different settings of epsilon on GraphSAGE+Ogbn-Arxiv: (1) $\epsilon \sim \mathcal{N}(0, \mathbf{I})$, (2) $\epsilon = 0$, and (3) $\epsilon = 0.2$, and report their performance under different sparsity levels in Table 15. We can see that trainable noisy parameters always bring the greatest performance gain to the model, which is consistent with previous practices in MoE that the randomness in the gating network is beneficial.

### G.4 SENSITIVITY ANALYSIS OF PARAMETER $p$

**How is $p$ determined in Equation (10)?** Since the egographs of individual nodes vary in size, we calculate $p$ proportionally as $p = \lceil r_p\% \cdot |\mathcal{V}(v_i)| \rceil$, where $r_p\%$ represents the ratio of selected columns in the ego-graph of $v_i$. In our experiments, we set $r_p\% = 50\%$ for simplicity.

**What is its impact on performance?** We conduct additional tests on GraphSAGE + OGBN-ARXIV using different values of $r_p\%$, as shown in Table 16. It can be observed that while excessively small values of $p$ negatively impact the performance of MoG, the model remains largely insensitive to variations in $p$ within a broader range.

Table 15: Ablation study on the noise control of the router network $\Psi$. $\epsilon \sim \mathcal{N}(0, \mathbf{I})$ corresponds to the original setting in our paper, $\epsilon = 0$ corresponds to completely remove the noise modeling, and $\epsilon = 0.2$ corresponds to fixing the noise coefficient.

| Sparsity | Train Acc | Valid Acc | Test Acc | $k$ |
|---|---|---|---|---|
| $\epsilon \sim \mathcal{N}(0, \mathbf{I})$ | | | | |
| 10% | 77.20 | 72.68 | 71.93 | 3 |
| 30% | 76.03 | 71.90 | 70.53 | 3 |
| 50% | 72.45 | 69.54 | 69.06 | 3 |
| $\epsilon = 0$ | | | | |
| 10% | 76.87 | 72.05 | 71.27 | 3 |
| 30% | 75.99 | 71.15 | 70.14 | 3 |
| 50% | 72.09 | 68.34 | 67.05 | 3 |
| $\epsilon = 0.2$ | | | | |
| 10% | 76.98 | 72.22 | 71.75 | 3 |
| 30% | 75.98 | 71.48 | 70.27 | 3 |
| 50% | 73.15 | 69.84 | 68.45 | 3 |

Table 16: The sensitivity analysis of $r_p\%$ on GraphSAGE+OGBN-ARXIV, with $s = 50$.

| $r_p\%$ | 0.1 | 0.3 | 0.5 | 0.7 |
|---|---|---|---|---|
| $s = 30$ | 69.92 | 70.03 | 70.53 | 70.41 |
| $s = 50$ | 68.27 | 69.12 | 69.06 | 69.19 |

### G.5 EXPERIMENTS INCORPORATING GLOBAL INFORMATION

While MoG prunes edges based on the node's local context, the edge importance evaluation function in Equation (9), $C^m(e_{ij}) = \text{FFN}(x_i, x_j, c^m(e_{ij}))$, can incorporate multi-hop or even global information. Specifically, we explored two approaches and conducted supplementary experiments accordingly:

1. **Expanding $x_v$ to $x_v^{(K)}$ with aggregated $K$-hop features:** To integrate multi-hop information, we expanded $x_v$ into

$$x_v^{(K)} = \text{CONCAT}(h_v^{(1)}, h_v^{(2)}, \ldots, h_v^{(K)}),$$

   where $h_v^{(k)} = \frac{1}{|N_k(v)|} \sum_{u \in N_k(v)} x_u$ represents the K-hop features for node $v$, with $N_k(v)$ denoting the K-hop neighbors of $v$.

2. **Incorporating $c^m(e_{ij})$ with prior global edge significance:** For the $c^m(e_{ij})$ function in Equation (9), we can consider global edge significance as prior guidance. This involved computing edge importance metrics such as PageRank, Betweenness Centrality, or Eigenvector Centrality across the entire graph and passing them to each ego graph to improve edge evaluation.

Table 17: Performance of MoG-$K$hop with multi-hop features and MoG-$m$ incorporating global edge significance as prior guidance on GraphSAGE+OGBN-ARXIV (node classification, 50% sparsity) and DeeperGCN+OGBG-PPA (graph classification, 50% sparsity).

| Method | GraphSAGE+OGBN-ARXIV | DeeperGCN+OGBG-PPA |
|---|---|---|
| MoG-1hop | 69.06 | 75.23 |
| MoG-2hop | 69.54 | 75.79 |
| MoG-3hop | 69.27 | 76.38 |
| MoG-PageRank | 69.03 | 75.70 |
| MoG-Betweenness | 69.14 | 75.68 |
| MoG-Eigenvector | 68.74 | 75.14 |

The results in Table 17 showcase that the performance gain from integrating global information varies across different tasks: in tasks such as graph classification, which rely more heavily on global information, MoG-3hop results in a notable $1.15\%$ accuracy improvement compared to MoG-1hop.

However, in node classification tasks, the gains from both hop expansion and global priors are relatively limited.

### G.6    ADDITIONAL EXPERIMENT WITH LINK PREDICTION

Table 18: Additional experiments on DeeperGCN+OGBL-COLLAB. The reported metrics represent the average of five runs. (Metric = Hits@50, Baseline = 53.53%)

| Sparsity % | 10 | 30 | 50 |
|---|---|---|---|
| Random | 52.92 | 47.38 | 44.62 |
| Local Degree | 53.57 | 51.80 | 49.92 |
| UGS | 53.65 | 52.25 | 49.67 |
| AdaGLT | 53.82 | 53.61 | 52.69 |
| MoG | 53.80 | 53.77 | 53.28 |

Table 19: Additional Experiments on GIN+PUBMED. The reported metrics represent the average of five runs. (Metric = ROC-AUC, Baseline = 0.895)

| Sparsity % | 10 | 30 | 50 |
|---|---|---|---|
| Random | 0.889 | 0.850 | 0.817 |
| Local Degree | 0.905 | 0.875 | 0.846 |
| UGS | 0.902 | 0.862 | 0.839 |
| AdaGLT | 0.898 | 0.884 | 0.851 |
| MoG | 0.910 | 0.893 | 0.862 |

We follow the experimental setting in Chen et al. (2021) to test link prediction on PUBMED and OGBL-COLLAB From Tables 18 and 19, we can further verify that MoG is also effective in the link prediction task, demonstrating strong generalization across diverse datasets and backbones.

## H    BROADER IMPACT AND DISCUSSION

### H.1    BROADER IMPACT

MoG, as a novel concept in graph sparsification, holds vast potential for general application. It allows for the sparsification of each node based on its specific circumstances, making it well-suited to meet the demands of complex real-world scenarios such as financial fraud detection and online recommender systems, which require customized approaches. More importantly, MoG provides a selectable pool for future sparsification, enabling various pruning algorithms to collaborate and enhance the representational capabilities of graphs.

### H.2    KEY CONTRIBUTIONS OF MoG

Briefly put, our contributions can be summarized as:

- **Node-Granular Customization**: We propose a new paradigm of graph sparsification by introducing, for the first time, a method that customizes both sparsity levels and criteria for individual node modeling based on their local context.
- **MoE for Graph Sparsification**: We design an innovative and highly pluggable graph sparsifier, dubbed Mixture of Graphs (MoG), which pioneers the application of Mixture-of-Experts (MoE) in graph sparsification, supported by a robust theoretical foundation rooted in the Grassmann manifold.
- **Empirical Evidence**: Our extensive experiments on seven datasets and six backbones demonstrate that MoG is **(1) a superior graph sparsifier**, maintaining GNN performance losslessly at 8.67% - 50.85% sparsity levels; **(2) a computational accelerator**, achieving a tangible 1.47 - 2.62× inference speedup; **(3) a performance booster**, which boosts ROC-AUC by 1.81% on OGBG-MOLHIV, 1.02% on OGBN-PROTEINS.

## H.3 Real-world Use Cases

SpMM frequently represents the primary computational bottleneck in GNNs, accounting for 50%–70% of the total computational load (Liu et al., 2023b), thereby severely limiting their scalability on large graphs. There are some real use cases:

- **Online Fraud Detection:** Financial transaction graphs are often massive, incurring significant SpMM costs during inference. The high computational cost of SpMM hinders the rapid detection and response required for financial fraud prevention, compromising user security.

- **Recommender Systems:** Large-scale user-item interaction graphs in recommender systems face substantial SpMM costs, restricting GNN deployment. MoG enhances the feasibility and responsiveness of GNNs in recommender systems.

- **Network Architecture Search (NAS):** NAS involves optimizing parameters for each architecture. The large parameter and gradient storage requirements of GNNs intensify memory demands. Moreover, NAS evaluates numerous candidate architectures, each requiring multiple training and validation cycles, amplifying SpMM's computational cost.

- **Federated Graph Learning:** The large-scale graph structures in Federated Graph Learning entail significant local SpMM computations and parameter transmission. MoG mitigates this by extracting sparse structures from local GNNs, reducing the parameter load sent to the central server.

We believe MoG's potential to alleviate SpMM bottlenecks could unlock broader applications of GNNs across these domains.

## I Post-sparsified Ego-graphs Assembling

we provide a detailed explanation of how post-sparsified ego-graphs are assembled as follows:

$$\widehat{\mathcal{G}} \leftarrow \{ \left| sign\left( \mathrm{TopK}(\widehat{\mathbf{A}}, |\mathcal{E}| \times s\%) \right) \right|, \mathbf{X} \}, \; s\% = \frac{1}{|\mathcal{V}|} \sum_{i=1}^{|\mathcal{V}|} s^{(i)}\%, \widehat{\mathbf{A}} = \sum_{i=1}^{|\mathcal{V}|} f\left( \widehat{\mathbf{A}}^{(i)} \right),$$

where $f : \mathbb{R}^{|\mathcal{N}(v)| \times |\mathcal{N}(v)|} \to \mathbb{R}^{|\mathcal{V}| \times |\mathcal{V}|}$ denotes the mapping of edges from the ego-graph to the global graph. After generating the post-sparsified ego-graphs, we compute the global sparsity $s\%$ by averaging the sparsity levels of each ego-graph and applying a function $f$ that maps each ego-graph into the global graph. Afterward, we sum the weights of unpruned edges across all ego-graphs to form the weighted adjacency matrix $\widehat{\mathbf{A}}$. Finally, we prune the global graph to achieve the target sparsity $s\%$, yielding the final sparsified graph.

