# OpenReview forum: "Graph Sparsification via Mixture of Graphs"
_ICLR.cc/2025/Conference — ICLR 2025 Spotlight_

### Official Review · Reviewer_aseP · 2024-11-02

**Soundness:** 4
**Presentation:** 4
**Contribution:** 2
**Rating:** 8
**Confidence:** 2

**Summary:**

This work leverages the Mixture-of-Experts (MoE) approach, a well-established technique in deep learning domains such as language modeling, to scale model capacity (i.e., the number of parameters) while managing computational demands. In this context, the MoE concept is applied dynamically and locally sparsity to the underlying graph during GNN inference. The authors demonstrate that this method maintains performance while reducing inference time.

**Strengths:**

I thought the writing was clear for the most part, with the exception being the contributions/intro (see Weaknesses). The figures are excellent. I haven't seen prior work on MoE re: sparsity, so I believe it to be original, although not very confident.

&nbsp;

This work could be significant. Indeed, GNN inference cost has motivated much GNN research in the past; see Questions for more of my thoughts on this.

**Weaknesses:**

**Making Contributions Clear**

I suggest making more clear the contributions of the paper and what the authors are claiming novelty on. Many papers use bullet points at the end of the introduction section to denote this; this work has bullet points near the end of the introduction, but they do not correspond to contributions, making the reading a bit confusing for regular readers of ICLR proceedings.

&nbsp;

**Some Missing Background in Graph Sparsification**

There is extensive literature on graph sparsification coming from the statistics and optimization world, commonly referred to under the umbrella term ‘Graph Structure Learning’. Most of these approaches indeed use some global sparsity promoting criteria; you could state how your approach has advantages over this more coarse approach. I include references to a classical work and a recent work. The latter has an overview of recent Graph Structure Learning approaches, most of which indeed use some global sparsity criterion in their objective.

&nbsp;

Friedman, Jerome, Trevor Hastie, and Robert Tibshirani. "Sparse inverse covariance estimation with the graphical lasso." Biostatistics 9.3 (2008): 432-441.

Wasserman, Max, and Gonzalo Mateos. "Graph Structure Learning with Interpretable Bayesian Neural Networks." Transactions on machine learning research (2024).

&nbsp;

**Missing Experiment/Ablation Study**

A simple baseline I expect to see would be to somehow prune the graph to a particular sparsity level and run a GNN on that. Perhaps some sort of edge sampling procedure selects a subset of edges to prune such that the global sparsity is set to ~=x, perhaps preferring edges to remove in proportional to the adjacent nodes' degree, while ensuring no disconnected nodes. Do so for a few sparsity levels. This would provide a good idea for how much benefit is really to be gotten from this whole effort, beyond the straightforward pruning approach.

Perhaps this ablation study is done implicitly in one of the baselines. Please correct me if so.

**Questions:**

**Significance**

Is there a real use case the authors can point to where SpMM compute limits people who are using/want to use GNNs? In my opinion, the significance of this type of ML research is measured by the significance of the problem it is solving/the capability it unlocks. Otherwise this is a more of a neat contribution to the literature which we simply speculate to be useful eventually.

&nbsp;

**The use of Gaussian Noise in the MoE step**

Perhaps I show my naivety re: MoE, but why do we believe added \epsilon \sim N(0,1) noise is of a suitable scale to effectively alter the routing? Is there a particular normalization of activations to ensure \epsilon interacts with numbers of this scale?

---

> ### Author Response · Authors · 2024-11-19
> **[Part 1/2] Response to Reviewer aseP**
>
> We sincerely thank you for your insightful comments and thorough understanding of our paper! Here we give point-by-point responses to your comments and describe the revisions we made to address them.
>
> ------
>
> > **Weaknesses 1**: Making Contributions Clear
>
> Following your suggestion, we have added a dedicated section in `Appendix H.1` to present our key contributions in clear and concise bullet points:
>
> * **Node-Granular Customization**: We propose a new paradigm of graph sparsification by introducing, for the first time, a method that customizes both sparsity levels and criteria for individual node modeling based on their local context.
> * **MoE for Graph Sparsification**: We design an innovative and highly pluggable graph sparsifier, dubbed Mixture of Graphs (MoG), which pioneers the application of Mixture-of-Experts (MoE) in graph sparsification, supported by a robust theoretical foundation rooted in the Grassmann manifold.
> * **Empirical Evidence**: Our extensive experiments on seven datasets and six backbones demonstrate that MoG is **(1) a superior graph sparsifier**, maintaining GNN performance losslessly at 8.67% - 50.85% sparsity levels; **(2) a computational accelerator**, achieving a tangible 1.47 - 2.62× inference speedup; **(3) a performance booster**, whichs boost ROC-AUC by 1.81% on OGBG-Molhiv and 1.02% on OGBN-Proteins;
>
>
> We hope that this addition can provide a clearer overview and emphasizes the novelty of our work.
>
> ------
>
> > **Weaknesses 2**: Some Missing Background in Graph Sparsification
>
> We appreciate the importance of situating our work within the broader context, including graph structure learning (GSL). In response, **we have appropriately cited the references you provided** and incorporated a discussion of the relevant works and their aspects in the revised manuscript under `Appendix H.2`. The key advantages of our approach over existing methods are as follows:
>
> 1. **Seamless Integration**: Unlike GSL methods, which often operate dependently of the backbone, MoG can be seamlessly embedded into various downstream graph tasks and GNN models. It functions as an effective sparsifier, inference accelerator, and performance enhancer without disrupting existing workflows.
> 2. **Dynamic Local Adaptation**: GSL methods typically rely on coarse-grained global metrics to evaluate topology importance and perform structure optimization. In contrast, MoG dynamically constructs sparse ego-graphs based on unique local contexts, thereby enhancing the quality and performance of the sparsified graph.
> 3. **High Customizability**: GSL methods often utilize a single metric for structural refinement, whereas MoG offers extensive customizability. It allows practitioners to tailor both sparsity criteria and levels to meet specific application needs. For example, in financial fraud detection, heterophilic pruning criteria can be applied, while in recommendation systems, PageRank-based pruning can be employed.
> ------
>
> > **Weaknesses 3**: Missing Experiment/Ablation Study
>
> Thank you for suggesting an insightful baseline. We respectfully note that the idea you proposed closely resembles Local Degree [1], which we have already compared in our paper. The essence of Local Degree is to remove edges based on the node degrees of different nodes. To provide a clearer comparison between MoG and Local Degree, we have extracted the relevant experimental results from Table 1 in Section 4 and presented them as Table A. The results demonstrate that MoG consistently outperforms Local Degree across all datasets. We attribute this to MoG's consideration of a broader range of local context beyond just node degree, including node features, spectral information, gradients, and more.
>
> *Table A: Comparison of MoG and Local Degree on OGBN-ARXIV and OGBN-PROTEINS datasets using GraphSAGE backbone.*
>
> | Dataset | OGBN-ARXIV | OGBN-ARXIV | OGBN-ARXIV  | OGBN-PROTEINS | OGBN-PROTEINS | OGBN-PROTEINS |
> | :---: | :---: | :---: | :---: | :---: | :---: | :---: |
> | Sparsity (%) | 10 | 30 | 50 | 10 | 30 | 50 |
> | Local Degree | 68.94 | 67.01 | 65.58 | 76.20 | 76.15 | 75.59 |
> | MoG | 71.93 | 70.53 | 69.06 | 77.78 | 77.49 | 76.46 |

---

> > ### Author Response · Authors · 2024-11-19
> > **[Part 2/2] Response to Reviewer aseP**
> >
> > > **Question 1**: Is there a real use case the authors can point to where SpMM compute limits people who are using/want to use GNNs?
> >
> > We sincerely appreciate your insightful perspective on assessing the significance of a paper. SpMM frequently represents the primary computational bottleneck in GNNs, accounting for 50%–70% of the total computational load, thereby severely limiting their scalability on large graphs[2]. There are some real use case:
> >
> > - **Online Fraud Detection:** Financial transaction graphs are often massive, incurring significant SpMM costs during inference. The high computational cost of SpMM hinders the rapid detection and response required for financial fraud prevention, compromising user security.
> > - **Recommender Systems:** Large-scale user-item interaction graphs in recommender systems face substantial SpMM costs, restricting GNN deployment. MoG enhances the feasibility and responsiveness of GNNs in recommender systems.
> > - **Graph Network Architecture Search (NAS):** Graph NAS involves optimizing parameters for each architecture. The large parameter and gradient storage requirements of GNNs intensify memory demands. Moreover, NAS evaluates numerous candidate architectures, each requiring multiple training and validation cycles, amplifying SpMM's computational cost.
> > - **Federated Graph Learning (FGL):** The large-scale graph structures in FGL entail significant local SpMM computations and parameter transmission. MoG mitigates this by extracting sparse structures from local GNNs, reducing the parameter load sent to the central server.
> >
> > We believe MoG's potential to alleviate SpMM bottlenecks could unlock broader applications of GNNs across these domains.
> >
> > ------
> >
> > > **Question 2**: The use of Gaussian Noise in the MoE step
> >
> > Thank you for your constructive inquiry! The usage of Gaussian noise in this paper follows the classical settings established in prior MoE studies [3,4]. In section G.3 of our manuscript, We test three different settings of epsilon on GraphSAGE+Ogbn-Arxiv: (1) $\epsilon\sim\mathcal{N}(0,\mathbf{I})$, (2) $\epsilon=0$, and (3) $\epsilon=0.2$, and report their performance under different sparsity levels in Table 15. We can see that trainable noisy parameters always bring the greatest performance gain to the model, which is consistent with previous practices in MoE that the randomness in the gating network is beneficial. Moreover, we believe the inclusion of trainable parameters $W_g$ and $W_n$ ensures the noisy scores are consistently scaled for effective routing.
> >
> > -------
> > [1] Structure-preserving sparsification methods for social networks. SNAM 2016
> >
> > [2] Dspar: An embarrassingly simple strategy for efficient gnn training and inference via degree-based sparsification. TMLR 2023
> >
> > [3] Outrageously large neural networks: The sparsely-gated mixture-of-experts layer. ICLR 2017
> >
> > [4] Sparse moe with language guided routing for multilingual machine translation. ICLR 2024

---

> ### Author Response · Authors · 2024-11-25
> **Thank you & Looking forward to further discussion!**
>
> Dear Reviewer aseP,
>
> We deeply appreciate your dedication to engaging in author-reviewer discussions. Here, we have outlined your key concerns and our responses for enhanced communication:
>
> 1. **Making Contributions Clear** **`Weakness 1`** In response to your suggestion, we have included a dedicated section in Appendix H.1 that highlights our key contributions in clear and concise bullet points.
> 2. **Some Missing Background in Graph Sparsification** **`Weakness 2`** We have respectfully cited the references you provided and included relevant discussions and comparisons. We sincerely thank you for helping us strengthen the presentation of our work!
> 3. **Missing Experiment/Ablation Study** **`Weakness 3`** We are amazed by your keen academic intuition! Your suggestion aligns precisely with a classic graph sparsification baseline, Local Degree. We have included the relevant experiments accordingly.
>
> We deeply and truly admire your academic intuition, and thank you immensely for your dedication to the reviewing process! We sincerely hope this addresses your concerns and look forward to further discussions.
>
> Warm regards,
>
> Authors

---

### Official Review · Reviewer_D6Ca · 2024-11-03

**Soundness:** 3
**Presentation:** 3
**Contribution:** 3
**Rating:** 8
**Confidence:** 3

**Summary:**

The authors propose a mixture-of-graphs (MoG) approach inspired by mixture-of-experts for graph sparsification. Instead of having global/uniform pruning criteria, they create a dynamic strategy tailored to each node’s neighborhood. This is done by utilizing each node’s egograph to select a few sparsifiers (experts) from a larger pool. The outputs of selected sparsifiers are ensembled using Grassmann manifold theory to generate a single sparsified ego graph per node and are later re-combined to form the final graph. Their approach outperforms baselines across several datasets, maintaining performance at higher sparsity levels and sometimes even improving it due to reduction of noise after sparsification.

**Strengths:**

1. Tackles a topical and highly relevant problem in graph learning on large graphs.
2. Highly flexible, well-motivated approach that accounts for local node variations while determining optimal pruning strategy.
3. Ample experiments across baselines and datasets, replete with sensitivity analysis, ablation studies, and efficiency comparison.

**Weaknesses:**

Adding below-mentioned minor clarifications around methodology may be useful:

1. How are post-sparsified ego-graphs assembled? Is it possible that for two nodes $i$ and $j$, the post-sparsified ego graph of $i$ connects to $j$ but the post-sparsified ego graph of $j$ doesn’t connect to $i$? If yes, how is this handled?
2. How do we select $p$ in equation 10? What is its impact on performance?
3. What does $D$ in eq. 13 correspond to? Is it from the original ego graph of the given node? If yes, why do we use the same $D$? Would it not introduce some approximation error since it doesn’t correspond exactly to the learned laplacian?
4. The method seems entirely local. For some downstream tasks, it may be relevant to take the global graph structure into consideration, for example, to ensure the graph stays connected. How does the proposed approach tackle this?

**Questions:**

A couple of questions on output:

1. In figure 1 (Middle), we attribute edge pruning to different sparsifiers. Did we perform this qualitative analysis on an actual sparsified graph, or is it an imagined example to represent our hypothesis? Although not required to perform this if not done already, I was simply wondering if it was the case and was interested in knowing more about it.
2. What is the difference between observed average sparsity just after ensembling in eq. 13 and after the post-sparsification step of eq. 14? Curious to understand how much more the graph density changes after the optimization in eq 11.

---

> ### Author Response · Authors · 2024-11-19
> **[Part 1/3] Response to Reviewer D6Ca**
>
> Thank you for your insightful comments and questions on our manuscript! We have carefully considered each point and have made the necessary revisions and clarifications to address your concerns:
>
> -----------
> > **Weakness 1**: How are post-sparsified ego-graphs assembled? Is it possible that for two nodes and $i$ and $j$, the post-sparsified ego graph of $i$ and $j$ connects to $j$ but the post-sparsified ego graph of $j$ doesn’t connect to $i$? If yes, how is this handled?
>
> **How are post-sparsified ego-graphs assembled?** We provide a detailed explanation of how post-sparsified ego-graphs are assembled as follows:
> $$
> \widehat{\mathcal{G}} \leftarrow \\{|sign({TopK}(\widehat{\mathbf{A}},|\mathcal{E}|\times s\\%))|,\mathbf{X}\\},\\; s\\%=\frac{1}{|\mathcal{V}|}\sum\_{i=1}^{|\mathcal{V}|} s^{(i)}\\%,\widehat{\mathbf{A}} = \sum\_{i=1}^{|\mathcal{V}|} f(\widehat{\mathbf{A}}^{(i)}),$$
>
> where $f:\mathbb{R}^{|\mathcal{N}(v)|\times|\mathcal{N}(v)|} \rightarrow \mathbb{R}^{|\mathcal{V}|\times|\mathcal{V}|}$ denotes the mapping of edges from the ego-graph to the global graph. After generating the post-sparsified ego-graphs, we compute the global sparsity $s\%$ by averaging the sparsity levels of each ego-graph and applying a function $f$ that maps each ego-graph into the global graph. Afterward, we sum the weights of unpruned edges across all ego-graphs to form the weighted adjacency matrix $\widehat{\mathbf{A}}$. Finally, we prune the global graph to achieve the global sparsity $s\%$, yielding the final sparsified graph.
>
> **Is it possible that for two nodes and $i$ and $j$, the post-sparsified ego graph of $i$ and $j$ connects to $j$ but the post-sparsified ego graph of $j$ doesn’t connect to $i$?** Yes, it is indeed possible. In such scenarios, whether this edge is retained or removed ultimately depends on the final global pruning step stated above. We sincerely hope this clarifies your concern.
>
> -----------
> > **Weakness 2**: How do we select $p$ in equation 10? What is its impact on performance?
>
>
> **How is $p$ determined in Equation 10?** Since the ego-graphs of individual nodes vary in size, we calculate $p$ proportionally as $p = \lceil r_p\\% \cdot |\mathcal{V}(v_i)|\rceil$, where $r_p\\%$ represents the ratio of selected columns in the ego-graph of $v_i$. In our experiments, we set $r_p\% = 50\%$ for simplicity. This detail has been explicitly clarified in the updated manuscript.
>
> **What is its impact on performance?** To address your concerns, we conducted additional tests on GraphSAGE + OGBN-Arxiv using different values of $r_p\\%$, as shown in Table A. It can be observed that while excessively small values of $p$ negatively impact the performance of MoG, the model remains largely insensitive to variations in $p$ within a broader range.
>
> _Table A. The sensitivity analysis of $r_p\\%$ on GraphSAGE+OGBN-Arxiv, with $s=50$._
> |$r_p\%$|0.1|0.3|0.5|0.7|
> |-|-|-|-|-|
> |$s=30$|69.92|70.03|70.53|70.41|
> |$s=50$|68.27|69.12|69.06|69.19|

---

> ### Author Response · Authors · 2024-11-19
> **[Part 2/3] Response to Reviewer D6Ca**
>
> > **Weakness 3**: What does $D$ in eq. 13 correspond to? Would it not introduce some approximation error since it doesn’t correspond exactly to the learned laplacian?
>
> We are sorry for causing confusion! The reference to $D$ in Equation (13) was indeed a typo, and we have corrected it to $D^{(i)}$, which represents the degree matrix of the ego-graph for node $i$. Thank you immensely for your kind attention, and we sincerely hope this can address your concerns.
>
> -----------
> > **Weakness 4**: The method seems entirely local. How does the proposed approach tackle the tasks which need global info?
>
> Thank you for your insightful question, which has greatly helped us improve our work! We would like to clarify an important point: while MoG prunes edges based on the node’s local context, the edge importance evaluation function in Equation (9), $C^m(e_{ij}) = \operatorname{FFN}\left( x_i, x_j, c^m(e_{ij}) \right)$, can incorporate multi-hop or even global information. Specifically, we explored two easy-to-implement extensions and conducted supplementary experiments accordingly:
>
> 1. **Expanding $x_v$ to $x_v^{(K)}$ with aggregated K-hop features:**
>    To integrate multi-hop information, we expanded $x_v$ into $x_v^{(K)} = \text{CONCAT}(h_v^{(1)}, h_v^{(2)}, \dots, h_v^{(K)})$, where $h_v^{(k)} = \frac{1}{|N_k(v)|} \sum_{u \in N_k(v)} x_u$ represents the K-hop features for node $v$, with $N_k(v)$ denoting the K-hop neighbors of $v$.
>
> 2. **Incorporating $c^m(e_{ij})$ with prior global edge significance:**
>     For the $c^m(e_{ij})$ function in Eq. (9), we can consider global edge significance as prior guidance. This involved computing edge importance metrics such as PageRank, Betweenness Centrality, or Eigenvector Centrality across the entire graph and passing them to each ego graph to improve edge evaluation.
>
> *Table B: Performance of MoG-Khop with Multi-hop Features and MoG-m Incorporating Global Edge Significance as prior guidance on OGBN-Arxiv+GraphSAGE (node classification, 50% sparsity) and OGBN-PPA+DeeperGCN (graph classification, 50% sparsity).*
> | Method           | OGBN-Arxiv+GraphSAGE | OGBN-PPA+DeeperGCN|
> |-|-|-|
> | MoG-1hop         | 69.06    | 75.23|
> | MoG-2hop         | 69.54    | 75.79|
> | MoG-3hop         | 69.27    | 76.38|
> | MoG-PageRank     | 69.03    | 75.70|
> | MoG-Betweenness  | 69.14    | 75.68|
> | MoG-Eigenvector  | 68.74    | 75.14|
>
> As observed, the performance gain from integrating global information varies across different tasks: in tasks such as graph classification, which rely more heavily on global information, MoG-3hop results in a notable 1.15% accuracy improvement compared to MoG-1hop. However, in node classification tasks, the gains from both hop expansion and global priors are relatively limited. Nevertheless, we hope this addresses your concern, demonstrating that **MoG can perform effectively even in scenarios where global information is crucial.**
>
>
> -----------
> > **Question 1**: Did we perform this qualitative analysis on an actual sparsified graph, or is it an imagined example to represent our hypothesis?
>
> The example in Figure 1 (Middle) is hypothetical and designed to illustrate the motivation behind our method. Nevertheless, we are happy to introduce a qualitative analysis based on experiments conducted on real datasets, as shown in Table B.
>
> *Table B: Importance Scores of Different Criteria on Various Datasets.Experiments were conducted using the GraphSAGE with a sparsity level of 30%. $I_D$, $I_{JS}$, $I_{ER}$, and $I_{GM}$ denote the aggregated expert score proportions for Degree, Jaccard Similarity, ER, and Gradient Magnitude, respectively.*
>
> | Dataset  | Nodes     | Edges   | Average Degree | $I_D$ | $I_{JS}$ | $I_{ER}$ | $I_{GM}$ |
> |:---:|:---:|:---:|:---:|:---:|:---:|:---:|:---:|
> | OGBN-Arxiv | 169,343 | 1,166,243  | 6.89 | 0.37 | 0.18 | 0.15 | 0.30 |
> | OGBN-Products | 2,449,029 | 61,859,140 | 25.26 | 0.16 | 0.23 | 0.25 | 0.36 |
> | OGBN-Proteins | 132,534 | 39,561,252 | 298.50 | 0.14 | 0.21 | 0.24 | 0.41 |
>
> Our findings reveal significant differences in the distribution of expert scores across datasets. Notably, in OGBN-Arxiv, which has a low average node degree, the Degree criterion exhibits higher expert scores. Conversely, in highly connected datasets like OGBN-Products and OGBN-Proteins, the Gradient Magnitude criterion shows the highest expert scores.
>
> This suggests that in graphs with low node connectivity, preserving important edges often relies more heavily on degree-based criterion to maintain graph connectivity. In contrast, in highly connected graphs, retaining important edges depends more on semantic information, such as gradient magnitude, which is learned during the training process. These observations align with the design principles of most state-of-the-art graph pruning methods, which prioritize node weights and gradients in their frameworks [1,2].

---

> ### Author Response · Authors · 2024-11-19
> **[Part 3/3] Response to Reviewer D6Ca**
>
> > **Question 2**: What is the difference between observed average sparsity just after ensembling in eq. 13 and after the post-sparsification step of eq. 14? Curious to understand how much more the graph density changes after the optimization in eq 11.
>
>
> Thank you for your insightful question! Pleasde allow us to clarify how the ego-graph of each node evolves throughout the process. Following Eq. (13), $k$ sparse ego-graphs $\{\widehat{\mathcal{G}}^{(i)}\_{m}\}_{m=1}^k$ are aggregated into a single representation $\widehat{\mathcal{G}^{(i)}} = \\{\widehat{\mathbf{A}^{(i)}}, \mathbf{X}^{(i)}\\}$. At this stage, $\widehat{\mathbf{A}^{(i)}}$ is derived from the ensembled Laplacian $\widehat{\mathbf{L}^{(i)}}$, which is inherently dense, i.e., it satisfies $||\widehat{\mathbf{A}^{(i)}}||_0 = \mathcal{E}^{(i)}$. We wish to emphasize that although $\widehat{\mathbf{A}}^{(i)}$ is dense at this stage, it is weighted—the weights are optimized on the Grassmann manifold, serving as critical guidance for the following post-sparsification process.
>
>
> Subsequently, as described in Eq. (14), $\widehat{\mathcal{G}^{(i)}}$ undergoes further post-sparsification, adjusting to a sparsity level of $s^{(i)}\%$ based on the learned node connectivity strengths optimized on the Grassmann manifold.
>
>
>
> ------
> [1] Edge sparsification for graphs via meta-learning
>
> [2] Rigging the Lottery: Making All Tickets Winners

---

> ### Author Response · Authors · 2024-11-25
> **Thank you & Looking forward to further discussion!**
>
> Dear Reviewer D6Ca,
>
> We humbly appreciate your recognition and thoughtful feedback on our work! For a better understanding of our rebuttal and revision, we have summarized your key concerns and our responses as follows:
>
> 1. **Ensembling Post-Sparsified Ego-Graphs** **`Weakness 1`**
> We have provided a detailed explanation of how multiple ego-graphs are merged back into a single large graph, which inherently addresses the conflicts you mentioned.
> 2. **Details on Parameter $k$** **`Weakness 2`**
> We elaborated on how $p$ is determined and reported its sensitivity analysis.
> 3. **Can MoG Tackle Tasks Requiring Global Information?**  **`Weakness 4`**
> We proposed two straightforward methods to incorporate global information into MoG and evaluated their performance.
>
> For other issues not mentioned here, please refer to our detailed rebuttal response. We sincerely hope this addresses your concerns! We respectfully look forward to further discussion with you.
>
> Warm regards,
>
> Authors

---

### Official Review · Reviewer_XfnA · 2024-11-05

**Soundness:** 3
**Presentation:** 3
**Contribution:** 3
**Rating:** 5
**Confidence:** 4

**Summary:**

This paper proposes a method called **MoG**, which uses the technique of MoE to integrate multiple ego-net sparsifier strategies in GNNs.

First, for each ego-net, the authors introduce a simple noisy top-k gating mechanism as a routing module to select K sparsifier experts, each with different sparsity levels and sparsification strategies.

Then, using Grassmann manifold techniques, the authors combine these sparsified ego graphs, with objective function Eq.11 and its closed-form solution in Eq.12.

The experiments show that compared to other sparsification methods:
- In terms of accuracy, MoG has a slight advantage in two node classification datasets and a more noticeable effect in two graph classification datasets.
- In terms of balancing inference speed and accuracy, MoG achieves higher accuracy when reaching the same speedup.

The authors also conducted additional experiments, such as sensitivity analysis.

**Strengths:**

1. The proposed method has a certain degree of innovation, especially in the use of MoE and the approach to combining graphs.
2. The proposed method can be integrated with any framework.
3. The paper includes extensive experiments.

**Weaknesses:**

1. Eq.11 is a core objective, but it is quite heuristic. Moreover, after obtaining the combined graph with Eq.12, there is a post-sparsification operation. Does the final ensembled sparse Laplacian truly integrate the eigenvectors of multiple sparsified ego-net Laplacians as the authors hope? Regardless of the degree achieved, it would be helpful to see experimental evidence here.

2. There are only two node classification and two graph classification datasets, which is relatively few.

**Questions:**

Please check weakness 1.

---

> ### Author Response · Authors · 2024-11-19
> **[Part 1/1] Response to Reviewer XfnA**
>
> Thank you for your valuable feedback on our manuscript! We have taken your comments seriously and have made the necessary revisions and additions to address the concerns raised. Below is our point-by-point rebuttal:
>
> ----------------
> > **Weakness 1**: Eq.11 is heuristic. Does the final ensembled sparse Laplacian truly integrate the eigenvectors of multiple sparsified ego-net Laplacians as the authors hope? Regardless of the degree achieved, it would be helpful to see experimental evidence here.
>
> Thank you for your valuable suggestion! In response, we provide an illustrative case study in `Appendix C.1` of the updated manuscript. Specifically, we examine how the Grassmann manifold enhances graph sparsification for the ego-graph (of node 2458) from the Ogbn-Arxiv dataset. We compare the original ego-graph, the sparse ego-graphs generated by three different sparsifiers, and the ensembled and **post-sparsified** ego-graphs derived through simple averaging and Grassmann optimization, as depicted in Figure 5. Our results demonstrate that simple averaging followed by sparsification leads to eigenvalue distributions that significantly deviate from the original graph. Conversely, **the Grassmann ensembling method preserves the spectral properties of each graph view**, producing a sparse ego-graph with an eigenvalue distribution that closely resembles that of the original graph. We believe this is because the post-sparsification is computed based on $\widehat{\mathbf{A}}^{(i)}$, which is optimized on the Grassmann manifold and better preserves the spectral information of multiple graph candidates.
>
>
>
> ----------------
> > **Weakness 2**: There are only two node classification and two graph classification datasets, which is relatively few.
>
> Thank you for your insightful comment, which greatly enhances the quality of our paper! In response, we have included additional experiments on link prediction tasks, specifically using DeeperGCN with OGBN-COLLAB and GIN with Pubmed, as shown in Table A and Table B, respectively. The link prediction experimental settings follow [1].
>
> These additions allow us to thoroughly evaluate our method **across three major graph tasks**—graph classification, node classification, and link prediction. The experiments now **cover four sparsity levels, six backbones, and eight datasets**, offering a comprehensive assessment of our approach under various conditions.
>
> *Table A: Additional Experiments on DeeperGCN + OGBL-COLLAB. The reported metrics represent the average of Five runs. (Metric = Hits@50, Baseline = 53.53%)*
> | Sparsity % | 10 | 30 | 50 |
> | :---: | :---: | :---: | :---: |
> | Random | 52.92 | 47.38 | 44.62 |
> | Local Degree | 53.57 | 51.80 | 49.92 |
> | UGS | 53.65 | 52.25 | 49.67 |
> | AdaGLT | **53.82** | 53.61 | 52.69 |
> | MoG | 53.80 | **53.77** | **53.28** |
>
>
> *Table B: Additional Experiments on GIN + Pubmed. The reported metrics represent the average of Five runs. (Metric = ROC-AUC, Baseline = 0.895)*
> | Sparsity % | 10 | 30 | 50 |
> | :---: | :---: | :---: | :---: |
> | Random | 0.889 | 0.850 | 0.817 |
> | Local Degree | 0.905 | 0.875 | 0.846 |
> | UGS | 0.902 | 0.862 | 0.839 |
> | AdaGLT | 0.898 | 0.884 | 0.851 |
> | MoG | **0.910** | **0.893** | **0.862** |
>
> Through the experiments in Tables A and B, we further verified that MoG is also effective in the link prediction task, demonstrating strong generalization across diverse datasets and backbones.
>
> -------
> We hope that these revisions properly address your concerns, and we are more than glad to respond to any further questions!
>
> ---
> [1] A Unified Lottery Ticket Hypothesis for Graph Neural Networks. ICML 2021

---

> ### Author Response · Authors · 2024-11-25
> **Thank you & Looking forward to further discussion!**
>
> Dear Reviewer XfnA,
>
> We sincerely appreciate your high commendation and thorough feedback on our work! To aid in better understanding our rebuttal and revision, we have summarized your key concerns and our responses as follows:
>
> 1. **How is the Final Ensembled Sparse Laplacian useful?** **`Weakness 1`**
> As visualized in `Appendix C.1` of the updated manuscript, compared to simply averaging $K$ sparse ego-graphs, the use of $\widehat{\mathbf{A}^{(i)}}$, optimized on the Grassmann manifold, better approximates the eigenvalue distribution of the original ego-graph.
> 2. **Insufficient Experiments** **`Weakness 2`**
> Respectfully following your suggestion, we have added two additional link prediction tasks. Notably, MoG has now been extensively validated across **four sparsity levels, six backbones, and eight datasets**, and we are deeply grateful for your valuable feedback.
>
> Thank you very much for your dedication to the reviewing process. We sincerely hope this addresses your concerns and look forward to further discussions.
>
> Warm regards,
>
> Authors

---

### Official Review · Reviewer_f3PB · 2024-11-06

**Soundness:** 3
**Presentation:** 3
**Contribution:** 3
**Rating:** 8
**Confidence:** 4

**Summary:**

This paper presents a novel graph sparsification method, Mixture-of-Graphs (MoG), to optimize Graph Neural Networks (GNNs) for large-scale graphs. MoG dynamically selects node-specific sparsification levels and criteria, improving computational efficiency and performance.
Inspired by the Mixture-of-Experts framework, MoG employs multiple sparsifier experts, each with distinct sparsity settings and pruning criteria. This approach customizes edge pruning for each node, addressing limitations of previous global sparsification techniques.
MoG assembles sparse subgraphs on the Grassmann manifold, enhancing graph structure while preserving node connectivity. Extensive experiments across diverse datasets demonstrate MoG’s ability to achieve significant sparsity with minimal performance loss.
The authors validate MoG’s effectiveness through experiments on large-scale datasets. MoG achieves faster GNN inference and better performance in some cases. MoG’s flexible sparsification method shows potential for advancing GNN deployment in resource-limited environments.

-----------------------------------
After reading all the responses and the improved manuscript, I think this paper can be considered to be accepted. Thus, I raised my rating to 'accept', because there is no '7'.

**Strengths:**

1. The experiments show the MoG's adaptability across different graph learning tasks. They also show MoG’s ability to improve inference speed while maintaining or even boosting accuracy (up to 1.74% in some cases) demonstrates its practical benefits.
2. The paper introduces a novel method, Mixture-of-Graphs (MoG), which applies the Mixture-of-Experts (MoE) concept to graph sparsification. Unlike traditional sparsification methods with uniform criteria, MoG combines sparsity levels and pruning criterias to each node’s local context.
3. The paper clearly defines the MoG framework, including the sparsifier expert selection process, node-wise routing, and mixture on the Grassmann manifold. The mathematical framework is well-detailed, particularly in describing the sparsifier expert selection via the noisy top-k gating mechanism and the Grassmann manifold’s role in mixing sparse subgraphs.
4. MoG's ability to achieve significant sparsity without a substantial performance drop highlights its potential to improve GNN deployment in resource-limited environments. By effectively balancing computational efficiency and accuracy, MoG could effectively serve as a plugin to boost GNN performance, from real-time social network analysis to large-scale molecular data processing.

**Weaknesses:**

1. The paper does not clarify in “4 EXPERIMENTS” or “Appendix F.6” why the specific 12 combinations of sparsity levels and criteria were selected. Additionally, the variance of each row across combinations in Table 8 is minimal, raising questions about the distinctiveness of each sparsifier.
Furthermore, in “Appendix F.6”, different sparsity criteria are applied in different datasets without an explanation of the selection rationale.
To enhance experimental completeness, the authors might consider exploring a wider range of sparsity combinations with greater variance. Additionally, providing the reasoning behind dataset-specific combinations would help readers understand the adaptability of MoG across different graph structures.

2. The process of integrating sparse subgraphs on the Grassmann manifold, as outlined in Section 3.4, is mathematically dense and lacks intuitive explanation. While the theoretical basis is strong, the connection between the Grassmann manifold’s properties and its benefits for graph sparsification may not be immediately clear to all readers.

3. The terminology for sparsifiers and experts appears inconsistent across sections, which could lead to confusion. For instance, “sparsifier experts” and “experts” are used interchangeably.

Minor Comment:
4. In Section 2, there is a spelling error in the title “TECHNICAL BACKGOUND,” which should be “TECHNICAL BACKGROUND.” I recommend reviewing the document to ensure there are no similar spelling errors throughout.

**Questions:**

Please refer to the weaknesses above.

---

> ### Author Response · Authors · 2024-11-19
> **[Part 1/2] Response to Reviewer f3PB**
>
> Thank you for your thorough review and insightful comments on our manuscript! We have carefully considered your feedback and have made the following revisions and clarifications to address the raised concerns.
>
> --------------------
> > **Weakness 1.1:** Why select the specific 12 combinations of sparsity levels and criteria?
>
> In the manuscript, we select these sparsity criteria because they are representative and easy to compute. However, **we respectfully emphasize that the combinations of sparsity levels and criteria in MoG can be easily customized for practitioners**.
>
> Furthermore, we use twelve candidate experts because the MoG method achieves a better balance between computational load and performance in this setup. To illustrate this point, Table B illustrates the inference time and accuracy of MoG with varying candidate experts $K$. It can be observed that MoG achieves the highest performance at $K=12$ with acceptable additional per-epoch time.
>
> *Table B: Inference Time & Accuracy of MoG with varying candidate experts $K$ when applyng MoG to OGBN-PROTEINS+GraphSAGE with $k=2$.*
> | $K$ | Per-epoch Time (s) | Accuracy(%) |
> |:-:|:-:|:-:|
> | 3 | 18.19 | 76.10 |
> | 6 | 19.70 | 76.43 |
> | 12 | 20.83 | 76.98 |
> | 16 | 21.74 | 76.90 |
> | 20 | 23.22 | 77.02 |
>
> > **Weakness 1.2:** Additionally, the variance of each row across combinations in Table 8 is minimal, raising questions about the distinctiveness of each sparsifier.
>
>
> Thank you for your detailed review and thoughtful comment! We selected the sparsity combinations presented in Table 8 as the final configuration in the paper to ensure reproducibility and enable a fair comparison with other baseline methods. **However, we respectfully emphasize that this does not indicate any incompatibility of MoG with sparsity combinations exhibiting higher variance.** To substantiate this, we conducted additional experiments:
>
> We tested **a broader range of sparsity configurations with greater variance** on the GraphSAGE + OGBN-Arxiv dataset. As shown in Table C, while the sparsifiers in the third row demonstrate significantly higher variance compared to those in the first row, the resulting global sparsity remains largely consistent. This outcome arises from MoG's ability to dynamically adjust expert loads, allocating fewer resources to high-sparsity experts (e.g., $15\%$ for $(1 - s_3) = 0.2$), thereby balancing the overall sparsity. This observation leads us to conclude that, regardless of differing sparsity combinations, MoG is capable of adjusting the expert load dynamically to approximate a suitable sparsity level for the graph data.
>
>   We hope this additional experiment demonstrates MoG's customizability and broad adaptability.
>
> *Table C: Performance of different sparsity level combinations when applying MoG to OGBN-ARXIV with GraphSAGE. In the table, $s_i$ represents the sparsity of the $i$-th sparsifier, and $l_i$ denotes its load.*
>
> | Dataset | $1-s_1$ | $1-s_2$ | $1-s_3$ | $l_1$ | $l_2$ | $l_3$ | $1-s$ | Accuracy |
> |:-:|:-:|:-:|:-:|:-:|:-:|:-:|:-:|:-:|
> | OGBN-ARXIV | 0.8 | 0.7 | 0.5 | 0.43 | 0.36 | 0.21 | 0.70 | 70.53 |
> | OGBN-ARXIV | 0.9 | 0.7 | 0.35 | 0.45 | 0.38 | 0.17 | 0.72 | 70.65 |
> | OGBN-ARXIV | 1 | 0.6 | 0.2 | 0.42 | 0.43 | 0.15 | 0.66 | 70.41 |

---

> ### Author Response · Authors · 2024-11-19
> **[Part 2/2] Response to Reviewer f3PB**
>
> > **Weakness 1.3:** Different sparsity combinations are applied in different datasets without an explanation of the selection rationale.
>
> In the manuscript, we selected specific combinations of sparsity levels for different datasets to achieve the target sparsities of $\\{10\\%, 30\\%, 50\\%, 70\\%\\}$, which **allows for a fair comparison with other graph sparsification methods under the same sparsity conditions**. This is because, even with the same sparsity combination, MoG optimizes a distinct global sparsity for each dataset due to its high customizability. Therefore, we adjusted the sparsity combinations for different datasets to closely match the desired sparsity levels, which also guarantees the reproducibility of our paper.
>
>
> We would like to emphasize, however, that these configurations are fully customizable, allowing users to adjust them according to their specific requirements and data scenarios.
>
> --------------------
> > **Weakness 2:**  The process of integrating sparse subgraphs on the Grassmann manifold is mathematically dense and lacks intuitive explanation. While the theoretical basis is strong, the connection between the Grassmann manifold’s properties and its benefits for graph sparsification may not be immediately clear to all readers.
>
> Thank you for your thoughtful question! We aim to address your concerns from both theoretical and practical perspectives.
>
> **From a theoretical standpoint**, the Grassmann manifold is fundamentally tied to the construction of the first $p$ eigenvectors, ensuring the orthogonality of eigenvectors corresponding to distinct eigenvalues and the normalized norm property of each eigenvector. By incorporating these constraints into the optimization problem, the solution is guided towards a matrix that exhibits "eigenvector-like" characteristics, aligning naturally with the graph Laplacian structure used in our framework. While it is true, as you noted, that the ego-graph integrated by the Grassmann manifold is mathematically dense, its adjacency matrix $\widehat{\mathbf{A}}^{(i)}$ is weighted. **These weights are optimized on the Grassmann manifold, serving as a solid foundation for subsequent post-sparsification.** A straightforward way to test the necessity and effectiveness of the Grassmann manifold is to directly apply expert scores to perform a weighted average of the sparse ego-graphs, followed by post-sparsification, rather than employing Eq. (11). To validate the necessity of the Grassmann manifold, we provide an experimental comparison in the next paragraph.
>
> **From a practical standpoint**, we provide an illustrative case study in `Appendix C.1` of the updated manuscript. Specifically, we examine how the Grassmann manifold enhances graph sparsification for the ego-graph of a node (node 2458) from the Ogbn-Arxiv dataset. We compare the original ego-graph, the sparse ego-graphs generated by three different sparsifiers, and the ensembled ego-graphs derived through simple averaging and Grassmann optimization, as depicted in Figure 5. Our results demonstrate that simple averaging followed by sparsification leads to eigenvalue distributions that significantly deviate from the original graph. Conversely, **the Grassmann ensembling method preserves the spectral properties of each graph view**, producing a sparse ego-graph with an eigenvalue distribution that closely resembles that of the original graph.
>
> In summary, we respectfully argue that employing the Grassmann manifold for sparse graph construction provides both theoretical and practical advantages by preserving the spectral properties of the graph, enabling more informed and effective graph sparsification.
>
> --------------------
> > **Weakness 3:**  The terminology for sparsifiers and experts appears inconsistent across sections.
>
> Thank you for wisely pointing this out! In MoG, the term "expert" is equivalent to "sparsifier expert", as illustrated in the third subfigure of Figure 2. In some instances, to emphasize the functionality of the "expert", we also refer to it as the "sparsifier expert".
>
>
> In our manuscript, "sparsifier" may refer to the "sparsifier expert of MoG", "ego-graph sparsifier" or "full-graph sparsifier", which could lead to ambiguity. To avoid this confusion, we have consistently referred to the sparsifier expert of MoG as "sparsifier expert" or "expert" in the revised manuscript.
>
> --------------------
> > **Weakness 4**:  Typos in the paper
>
> Thank you for your detailed review and for kindly pointing out the spelling errors in our manuscript. We have addressed the issues you mentioned and conducted a thorough review of the entire document to identify and correct additional errors, such as changing "paramter" to "parameter" in Section 4.3 (obs5) and "NETWOR" to "NETWORK" in the title of G.3. These corrections have been incorporated into the revised manuscript.

---

> ### Author Response · Authors · 2024-11-25
> **Thank you & Looking forward to further discussion!**
>
> Dear Reviewer f3PB,
>
> We would like to extend heartfelt thanks to you for your time and efforts in the engagement of the author-reviewer discussion. To facilitate better understanding of our rebuttal and revision, we hereby summarize your key concerns and our responses as follows:
>
> 1. **Explanation of MoG's Sparsity Combination** **`Weakness 1`**
> We respectfully clarify that (1) $K=12$ represents the optimal trade-off between performance and computational efficiency for MoG, and (2) MoG is inherently adaptable to sparsity combinations with greater variance.
> 2. **Variation of Sparsity Combination Across Datasets** **`Weakness 1`**
> This variation arises from the need to tune MoG to achieve the desired global sparsity, enabling fair comparisons with other graph sparsification methods under similar sparsity levels.
> 3. **How the Grassmann Manifold Benefits Graph Sparsification** **`Weakness 2`**
> The weighted ego-net adjacency matrix $\widehat{\mathbf{A}^{(i)}}$ is optimized on the Grassmann manifold, efficiently preserving the spectral properties of each graph view and effectively informing the subsequent post-sparsification procedure.
>
> For other issues not mentioned here, please refer to our detailed rebuttal response. We sincerely hope this addresses your concerns! We respectfully look forward to further discussion with you.
>
> Warm regards,
>
> Authors

---

> ### Author Response · Authors · 2024-11-30
> **Thank you immensely!**
>
> We extend our heartfelt thanks to `Reviewer f3PB` for their increased support of our paper! We are pleased that our rebuttal have sufficiently addressed your concerns.

---

### Author Response · Authors · 2024-11-19
**Summary of Manuscript Revision**

Thank you to all the reviewers for your thoughtful and constructive comments! We are really encouraged to see that the reviewers appreciate some positive aspects of our paper, such as technical novelty (Reviewers `XfnA`, `aseP`, `f3PB`), theoretical guarantees (Reviewer `f3PB`), thorough experimental validation (Reviewers `D6Ca`, `XfnA`) and practical benefits(Reviewers `aseP`, `f3PB`).

Your expertise significantly helps us strengthen our manuscript – this might be the most helpful review we have received in years! In addition to addressing your thoughtful comments point-to-point on OpenReview forum, we have made the following modifications to the newly uploaded manuscript (all updated text is highlighted in blue):

* **Intuitive Explanation of Grassmann Manifold:** We have further discussed and visualized the impact of Grassmann manifold ensembling on the eigenvalue distribution in `Appendix C.1`.
* **Incorporating Global Information:** We have explored and tested two approaches to enable MoG to perceive multi-hop and global information, as detailed in `Appendix G.5`.
* **Link Prediction Experiment:** We have comprehensively supplemented the experiments on link prediction tasks in `Appendix G.6`.
* **Additional Analysis of Parameter $p$:** We have provided detailed discussion on parameter $p$ in `Appendix G.4`.
* **Other revisions:** We have added relevant works and highlighted our contributions in `Appendix H`, and carefully corrected typos throughout the manuscript.

We have made earnest efforts to address the primary concerns raised. We also respectfully look forward to the thoughtful feedback from the reviewers to further enhance the quality of our manuscript.

---

### Author Response · Authors · 2024-11-22
**General Response**

Dear Reviewers,

Thank you for your thorough and insightful reviews. We sincerely appreciate your feedback, which has significantly enhanced our paper! Below, we summarize the key concerns raised and our corresponding responses:

- **How Grassmann manifold benefits MoG** (`Reviewer f3PB, XfnA`)
  We have provided both theoretical and experimental explanations on how Grassmann ensembling effectively preserves the spectral properties of multiple graph views prior to post-sparsification.
- **Implementation details of MoG** (`Reviewers f3PB, D6Ca`)
  We have clarified (1) the rationale behind MoG’s sparsity combination and (2) the process of post-sparsifying ego-graphs.
- **Additional experiments** (`Reviewer XfnA, D6Ca`)
  We have included experiments on link prediction tasks and scenarios requiring global information, extending MoG's evaluation to *a total of eight datasets*.
- **Ablation and sensitivity studies** (`Reviewers D6Ca, aseP`)
  We have added a sensitivity analysis for the parameter $p$ and a comparison with a Local Degree-style baseline.

Once again, we are truly grateful for your valuable feedback and are happy to address any further concerns or questions!

Sincerely,

Authors

---

### Meta-Review · Area_Chair_p7QS · 2024-12-18

**Metareview:**

In this submission, the authors leverage the idea of the mixture of graphs (MoG) to achieve adaptive graph sparsification. Applying MoG to GNNs can reduce the computational costs significantly while maintaining high performance on large-scale graph learning tasks. The idea of MoG is simple but effective, which works for various tasks and datasets. In the revised paper and the rebuttal, the authors provide sufficient experiments and detailed analytic content to verify the rationality and feasibility of the proposed method.

**Additional Comments On Reviewer Discussion:**

In the rebuttal phase, two reviewers interacted with the authors and increased their scores. Most of the reviewers are satisfied with the authors' rebuttals. Although one reviewer scored this work negatively, AC finally has decided to accept this work after reading the submission, the comments, and the rebuttals.

---

### Decision · Program_Chairs · 2025-01-22

Accept (Spotlight)